# The Effect of Low Irradiance on Leaf Nitrogen Allocation and Mesophyll Conductance to CO_2_ in Seedlings of Four Tree Species in Subtropical China

**DOI:** 10.3390/plants10102213

**Published:** 2021-10-18

**Authors:** Jingchao Tang, Baodi Sun, Ruimei Cheng, Zuomin Shi, Da Luo, Shirong Liu, Mauro Centritto

**Affiliations:** 1School of Environmental and Municipal Engineering, Qingdao University of Technology, Qingdao 266525, China; tjc@qut.edu.cn (J.T.); sunbaodi0927@126.com (B.S.); 2Key Laboratory of Forest Ecology and Environment of National Forestry and Grassland Administration, Research Institute of Forest Ecology, Environment and Protection, Chinese Academy of Forestry, Beijing 100091, China; chengrm@caf.ac.cn (R.C.); luoda2010@163.com (D.L.); liusr@caf.ac.cn (S.L.); 3Co-Innovation Center for Sustainable Forestry in Southern China, Nanjing Forestry University, Nanjing 210037, China; 4Institute for Sustainable Pant Protection, National Research Council of Italy, Strada delle Cacce 73, 10135 Torino, Italy; mauro.centritto@cnr.it

**Keywords:** leaf nitrogen allocation, mesophyll conductance, photosynthetic nitrogen use efficiency, low irradiance, N-fixing tree species

## Abstract

Low light intensity can lead to a decrease in photosynthetic capacity. However, could N-fixing species with higher leaf N contents mitigate the effects of low light? Here, we exposed seedlings of *Dalbergia odorifera* and *Erythrophleum fordii* (N-fixing trees), and *Castanopsis hystrix* and *Betula alnoides* (non-N-fixing trees) to three irradiance treatments (100%, 40%, and 10% sunlight) to investigate the effects of low irradiance on leaf structure, leaf N allocation strategy, and photosynthetic physiological parameters in the seedlings. Low irradiance decreased the leaf mass per unit area, leaf N content per unit area (*N*_area_), maximum carboxylation rate (*V*_c__max_), maximum electron transport rate (*J*_max_), light compensation point, and light saturation point, and increased the N allocation proportion of light-harvesting components in all species. The studied tree seedlings changed their leaf structures, leaf N allocation strategy, and photosynthetic physiological parameters to adapt to low-light environments. N-fixing plants had a higher photosynthesis rate, *N*_area_, *V*_cmax_, and *J*_max_ than non-N-fixing species under low irradiance and had a greater advantage in maintaining their photosynthetic rate under low-radiation conditions, such as under an understory canopy, in a forest gap, or when mixed with other species.

## 1. Introduction

Radiation is a source of energy for plants. Through photosynthesis, green plants use light to synthesize carbohydrates from water and CO_2_, which are necessary for maintaining growth and development. The low radiation conditions in the understory canopy of subtropical forests affect the survival and growth of forest tree seedlings [1]. Low light intensity can lead to a decrease in photosynthetic capacity, forcing plants to change their leaf photosynthesis system and structure to increase their light-harvesting ability [2,3,4,5]. Under low irradiance, plants usually adjust their leaf nitrogen (N) allocation strategies, such as increasing the fraction of leaf nitrogen (N) allocated to light-harvesting (*P*_L_) [5,6,7,8], and some plants may also change the fraction of leaf N allocated to Rubisco (*P*_R_) and bioenergetics (*P*_B_) to balance the light reaction with carbon assimilation and achieve optimal photosynthetic efficiency [8,9]. However, some plants do not adjust their *P*_R_ and *P*_B_ [10], which may be because some plants store many compounds containing N, such as free amino acids [11], inorganic N (NO_3_^−^, NH_4_^+^) [12], and some inactive Rubisco [12,13], and allocate these N sources to light-harvesting systems under low light levels.

Under low irradiance, the leaf thickness may decrease and leaf area may increase, resulting in a lower leaf mass per unit area (LMA) [1,4,14], which increases the area receiving light [8]. Low irradiance could also result in a reduction in the surface area of mesophyll cells per unit leaf area, as well as a smaller area of mesophyll cells through which CO_2_ can diffuse into the chlorophyll [15,16]. These changes subsequently affect the mesophyll conductance to CO_2_ (*g*_m_) and, in turn, affect the CO_2_ concentration in chloroplasts (*C*_c_) [17,18]. Low irradiance decreased *g*_m_ [19,20] or did not significantly affect *g*_m_ in different species [21,22]. Therefore, changes in *g*_m_ in different species should be further studied.

The allocation of N in photosynthetic systems and *g*_m_ are common and important factors affecting photosynthetic N use efficiency (PNUE) [23,24], which is the ratio of the photosynthetic rate to the leaf N content [25,26] and reflects the N resources used for photosynthesis, an important leaf trait. Many authors have studied the PNUE of various plants under changing light intensities, and the changes in the PNUE of different plant species under different light intensities were inconsistent; some studies found that, under low irradiance, the PNUE increased [5,20,27], while others found that it decreased [28] or remained unchanged [7,29]. However, few studies have been conducted on whether low-irradiance treatment can affect the PNUE of N-fixing trees and the relevant internal control mechanisms of leaf N allocation and *g*_m_. We suspect that N-fixing species with sufficient N in their leaves could increase their *P*_R_, *P*_B_, and *P*_L_ to increase the PNUE under low-irradiance treatment, and maintain photosynthetic capacity and growth better than non-N-fixing species under low-irradiance environments. 

In this study, we exposed *Dalbergia odorifera* and *Erythrophleum fordii* (N-fixing trees), and *Castanopsis hystrix* and *Betula alnoides* (non-N-fixing trees) seedlings to three levels of irradiance (100%, 40%, and 10% sunlight irradiance) and estimated their photosynthesis, PNUE, leaf N allocation, and *g*_m_ values. These species are locally vital broad-leaved trees with high economic value, which are commonly used to change *Pinus massoniana* and *Cunninghamia lanceolata* pure forests into mixed broadleaf-conifer forests or mixed broad-leaved forests. This requires the selected species to be planted in forest gaps, mixed with other species, or directly on bare ground; therefore, their tolerance under low light conditions (e.g., in the understory canopy) will affect their survival and growth. Full light conditions of 10% and 40% are common in forest gaps as well as with mixed planting conditions, while 100% light conditions are typical for direct planting on bare land.

The aim of this study was to (1) determine the effects of low irradiance on leaf structure, leaf N allocation strategy, and photosynthetic physiological parameters (e.g., *g*_s_, *g*_m_, and photosynthetic rate) and (2) evaluate whether N-fixing plants are better able to maintain their photosynthetic rate under low-radiation conditions compared to non-fixing plants.

## 2. Results

*N*_area_ and *N*_mass_ in *D. odorifera* and *E. fordii* seedling leaves were significantly higher than those in *C. hystrix* and *B.*
*alnoides* under each irradiance treatment (Table 1). There was a significant decrease in *N*_area_ and the LMA of all four species under the 10% and 40% irradiance treatments when compared with the 100% treatment (Table 1). *A*_sat_ of *E. fordii* under the 40% irradiance treatment was significantly higher than that under the other treatments; however, *A*_sat_ of *C. hystrix* under the 10% and 40% irradiance treatments, and *A*_sat_ of *B. alnoides* under the 10% irradiance treatment were significantly lower than that under the 100% treatment (Table 1). *N*_mass_ of *E. fordii*
*C. hystrix* and *B.*
*alnoides* was significantly higher under the 10% irradiance treatment than that under the 100% treatment (+25.6%, +33.8%, and +23.6%, respectively; Table 1). The PNUE_sat_ of *D. odorifera* under the 10% irradiance treatment, and of *E. fordii* under the 10% and 40% irradiance treatments were significantly higher than that under the 100% treatment; however, the PNUE_sat_ of *C. hystrix* under the 10% and 40% irradiance treatments was significantly lower than that under the 100% treatment (Table 1).

Both *g*_s_ and *g*_m_ of *C. hystrix* under the 10% and 40% irradiance treatments, and *g*_s_ and *g*_m_ of *B. alnoides* under the 10% irradiance treatment were significantly lower than those under the 100% treatment (Table 2). In contrast, *g*_s_ of *D. odorifera* under the 10% and 40% irradiance treatments were higher than that under the 100% treatment (+32.8% and +35.8, respectively), whereas *g*_m_ of *D. odorifera* under the 10% and 40% irradiance treatments was lower than that under the 100% treatment (−27.0% and −21.9%, respectively). *g*_m_ of *E. fordii* under the 40% irradiance treatment was significantly higher than that of the other treatments (Table 2). In *D. odorifera*, *C*_i_ under the 10% and 40% irradiance treatments, and *C*_c_ under the 10% irradiance treatment were higher than that under 100% irradiance, and in *E. fordii*, *C*_i_ under 10% irradiance treatment, and *C*_c_ under 10% and 40% irradiance treatments were higher than those under 100% irradiance (Table 2). Irradiance treatments did not significantly affect the CO_2_ drawdown (*C*_i_-*C*_c_) in any of the four tree species studied (Table 2).

*V*_cmax_ and *J*_max_ of *D. odorifera*, *E. fordii*, and *B. alnoides* under the 10% irradiance treatments, and *V*_cmax_ and *J*_max_ of *C. hystrix* under the 10% and 40% irradiance treatments were lower than those under 100% irradiance (Table 3). In contrast, *V*_cmax_ of *D. odorifera* and *V*_cmax_ and *J*_max_ of *E. fordii* under the 40% irradiance treatment were higher than those under 100% irradiance (Table 3).

In *D. odorifera*, *P*_R_, *P*_B_, *P*_L_, and *P*_P_ under the 10% and 40% irradiance treatments were higher than those under 100% irradiance, but *P*_Other_ under the 10% and 40% irradiance treatments were lower than those under 100% irradiance (Table 4). In *E. fordii*, *P*_R_, *P*_L_, and *P*_P_ under 10% and 40% irradiance treatments, and *P*_B_ under 40% irradiance treatment were higher than those under 100% irradiance. However, *P*_CW_ and *P*_Other_ under the 10% and 40% irradiance treatments were lower than those under 100% irradiance (Table 4). In *C. hystrix*, *P*_L_ under the 10% and 40% irradiance treatments, and *P*_CW_ under the 40% irradiance treatment were higher than those under 100% irradiance, but *P*_R_, *P*_B_, and *P*_Other_ under the 10% and 40% irradiance treatments were lower than those under 100% irradiance (Table 4). In *B. alnoides*, *P*_L_ under the 10% irradiance treatment, *P*_R_ and *P*_B_ under the 40% irradiance treatment, and *P*_P_ under the 10% and 40% irradiance treatments were higher than those under the 100% irradiance treatment, but *P*_CW_ under the 10% irradiance treatment was lower than that under 100% irradiance (−32.1%, Table 4).

The apparent quantum yield (AQY) and *R*_n_ of *D. odorifera*, *C. hystrix*, and *B. alnoides* were not significantly affected by low-irradiance treatment; however, the AQY of *E. fordii* under the 10% and 40% irradiance treatments were significantly higher than that under 100% irradiance, and *R*_n_ of *E. fordii* under the 10% and 40% irradiance treatments were significantly lower than those under 100% irradiance (Table 5). The light compensation point (LCP) of *D. odorifera* and *C. hystrix* under the 10% and 40% irradiance treatments, and *E. fordii* and *B. alnoides* under the 10% irradiance treatment were significantly lower than those under 100% irradiance (Table 5). The light saturation point (LSP) of *D. odorifera* and *E. fordii* under the 10% irradiance treatment, and *C. hystrix* and *B. alnoides* under the 10% and 40% irradiance treatments were significantly lower than those under 100% irradiance (Table 5).

*A*_100_ and *A*_400_ in *D. odorifera* and *E. fordii* seedling leaves were significantly higher than those in *C. hystrix* and *B.*
*alnoides* under 10% irradiance treatment (Table 6). *A*_100_, PNUE_100_ and PNUE_400_ of *D. odorifera* and *E. fordii* under the 10% and 40% irradiance treatments were significantly higher than that under the 100% treatment (Table 6). *A*_400_ of *E. fordii* under the 40% irradiance treatment was significantly higher than that under the other treatments (Table 6). *A*_100_, *A*_400_ and PNUE_400_ of *C. hystrix* under the 10% and 40% irradiance treatments were significantly lower than that under the 100% treatment (Table 2). *A*_400_ of *B.*
*alnoides* was significantly lower than that under the 40% and 100% treatments, but PNUE_100_ of *B.*
*alnoides* was significantly higher than that under the 100% treatment (Table 6). 

The PNUE_sat_ was significantly, linearly related to *P*_R_, *P*_B_, and *P*_P_ in all four tree species in all treatments (*p* < 0.001, Figure 1a,b,d). In contrast, the PNUE_sat_ of all four tree species was significantly positively related to *P*_L_ only under the 10% irradiance treatment (*p* < 0.001, Figure 1c). There was no significant positive correlation between PNUE_sat_ and *g*_m_ in these four species (Figure 2).

## 3. Discussion

Under the 10% and 40% irradiance treatments, plants consistently reduced their LMAs (Table 1), that is, they reduced their leaf thickness to improve the transmittance of light and increase the leaf area to increase the area receiving light [1,4,8,14]. Leaves may change their arrangements of mesophyll cells and chloroplasts to increase their light capture efficiency, which allows their light-harvesting capacity to be increased and sustain photosynthesis [3,4]. The *N*_mass_ of all four tree species included in this study was higher under the 10% irradiance treatment. Although the increase in *N*_mass_ in *D. odorifera* was not significant, it was significant in the other three species (Table 1). N is an important component of chlorophyll, and plants increase the concentration of N to increase chlorophyll synthesis under low irradiance [7,8,20]. As *N*_area_ = LMA × *N*_mass_, the significant decrease in LMA led to a decrease in *N*_are__a_ under the 10% and 40% irradiance treatments, indicating that thinner leaves had a lower concentration of N per unit area [1,10,30]. We hypothesized that N-fixing trees could fix nitrogen from the air; therefore, the reduction in *N*_area_ under low light may be smaller than that of non-N-fixing tree species. However, our results indicate that the decrease in the proportion of *N*_area_ was not lower in N-fixing trees than that in non-N-fixing trees (*D. odorifera:* −55.70%, *E. fordii:* −22.39%, *C. hystrix:* −22.54%, and *B. alnoides:* −45.63%). The N fixation capacities of *D. odorifera* and *E. fordii* did not limit the reduction in *N*_area_ under low light treatment. 

In this study, *A*_sat_ significantly reduced in two non-N-fixing tree species, *C. hystrix* under the 10% and 40% irradiance treatments, and *B. alnoides* under the 10% irradiance treatment (Table 1). Reduced *A*_sat_ under low light treatment has been observed in many other studies [20,31], and many researchers have reported that *C*_c_, *V*_cmax_, and *J*_max_ are important factors affecting *A*_sat_. CO_2_ is a key material for photosynthesis [32], and *V*_cmax_ and *J*_max_ are key biochemical parameters of the photosynthetic capacity [33]. In this study, the *C*_c_ of *C. hystrix* and *B. alnoides* did not change significantly from the 10% to 100% irradiance treatments (Table 2), but the *V*_c__max_ and *J*_max_ of *C. hystrix* under the 10% and 40% irradiance treatments, and *V*_c__max_ and *J*_max_ of *B. alnoides* under the 10% irradiance treatment were lower than those under 100% irradiance (Table 3), which were the main reasons for the reduction in *A*_sat_ in these two species (Table 1). Although *V*_c__max_ and *J*_max_ of two N-fixing tree species, *D. odorifera* and *E. fordii*, were reduced under 10% irradiance (Table 3), the *C*_c_ of these species was significantly increased under 10% irradiance (Table 2), which resulted in the absence of significant changes in *A*_sat_ (Table 1). The *A*_sat_, *V*_c__max_, and *J*_max_ of *E. fordii* were highest under 40% irradiance, suggesting that moderate shading may be more beneficial to its growth (Table 1 and Table 3).

*g*_m_ in *D. odorifera, C. hystrix,* and *B. alnoides* seedlings decreased under 10% irradiance, which was consistent with previous studies [19,20] (Table 2). *g*_m_ could be affected by leaf anatomical differences, such as cell wall thickness, surface area of mesophyll cells, number of mesophyll layers, and leaf stomata density [17,18]. Variations in LMA could be driven by several anatomical traits, such as the cell wall thickness and number of mesophyll layers [34], and changes in LMA always influence *g*_m_ [35]. If a lower LMA is the result of mesophyll cell wall thinning, it will increase *g*_m_ [36,37]; if it is the result of a lower number of mesophyll layers, it will decrease *g*_m_ [38]. In this study, the LMA of *D. odorifera*, *C. hystrix,* and *B. alnoides* decreased under the 10% irradiance treatment, indicating that low light may decrease the number of mesophyll layers in these tree seedlings. 

There was no significant change or increase in *A*_sat_, but *N*_area_ was significantly reduced in the two N-fixing tree species under 10% and 40% irradiance treatments, which caused an increase in the PNUE_sat_ in these trees. The PNUE_sat_ in *C. hystrix* decreased under 10% and 40% irradiance treatments (Table 1). The PNUE_sat_ of different tree species can respond differently to low light treatment, and may increase [5,20,27], decrease [28], or show no marked change [7,29]. This is related to the functional characteristics of different tree species. Many scholars have suggested that *P*_R_ and *P*_B_ are the main factors affecting PNUE_sat_ [39,40]. In this study, *P*_R_ and *P*_B_ were the main factors affecting the variation in the PNUE_sat_ under the 100% and 40% irradiance treatments; however, under the 10% irradiance treatment, the effects of *P*_L_ on PNUE_sat_ became significant, and the effects of *P*_R_ and *P*_B_ on PNUE_sat_ decreased (Figure 1). The ability to harvest light under low light treatment is a key factor limiting photosynthesis, and the importance of carboxylation and electron transport capacity decreases under such treatment, but persists [8,31]. We speculated that changes in *g*_m_ may affect PNUE_sat_, based on the role of N in mesophyll conductance [41,42]. However, our results indicated that the effect of *g*_m_ on PNUE_sat_ was not significant under varying light treatments in all tree species (Figure 2). 

As these species are commonly used to plant in gaps or mixed with other species, their tolerance to low light conditions will affect the growth effect after planting. All four species decreased the LMA to increase the area receiving light (Table 1) [1,4,8,14], increased *P*_L_ to increase their light-harvesting capacity and sustain photosynthesis (Table 4) [3,4], and decreased the LCP and LSP to increase the ability to use low light (Table 5) under low light conditions. Meanwhile, N-fixing plants exhibited some other adaptations to low light conditions, such as increased *A*_100_, PNUE_100_ and PNUE_400_ under the 10% and 40% irradiance treatments (Table 6). N-fixing plants also had higher *A*_100_, *A*_400_, *N*_area_, *V*_cmax_ and *J*_max_ than non-N-fixing species under the 10% irradiance treatment (Table 1, Table 3 and Table 6). Overall, these two N-fixing plant seedlings had higher photosynthetic rates, photosynthetic ability and higher adjustment ability of photosynthetic N use under low light conditions. AQY refers to the ability to use low light [43]. *E. Fordii* exhibited improved AQY under the 10% and 40% irradiance treatments, and also reduced *R*_n_ under the 10% and 40% irradiance treatments to reduce respiratory expenditure (Table 5). In conclusion, these results suggest that the adaptability of these two N-fixing species to low light environments is better than that of non-N-fixing species.

We previously studied the interspecific differences between *D. odorifera* and *E. fordii* (N-fixing trees), and *C. hystrix* and *B. alnoides* (non-N-fixing trees) [44], and how they are affected by soil N deficiencies [45]. The data obtained under high light intensity in this manuscript are the same as those used by Tang et al. [44] and the high nitrogen condition reported by Tang et al. [45], which were used as the “Control group.” In [44], N-fixing trees had higher *N*_area_ and *N*_mass_, but lower *P*_R_, *P*_B_, and PNUE than non-N-fixing trees. In [45], soil N deficiency had less influence on the leaf N concentration and photosynthetic ability in the two N-fixing trees. Combined with the results of this study, we consider that nitrogen-fixing plants are suitable species for afforestation, and could be independently planted in poor soil, mixed with non-N-fixing species, or planted in gaps.

The *P*_L_ of all four species increased to improve their light-trapping ability under low irradiance treatments (Table 4), which was consistent with previous studies [31,46]. However, different tree species employ different strategies to increase their *P*_L_: *D. odorifera* seedling leaves decreased *P*_Other_ to increase *P*_R_, *P*_B_, and *P*_L_; *E. fordii* seedling leaves decreased *P*_Other_ and *P*_CW_ to increase *P*_L_ and *P*_R_; *C. hystrix* seedling leaves decreased *P*_R_, *P*_B_, and *P*_Other_ to increase *P*_L_; and *B. alnoides* seedling leaves decreased *P*_CW_ to increase *P*_L_ under 10% irradiance (Table 4). Many studies have also observed changes in N allocation under low light treatment [4,5,8,9,47]. These different strategies are related to the ecological characteristics of each tree species, but the goal is the same (reducing some other N components and increase light-harvesting N components under low light treatment). However, why these tree species reduce the corresponding N components requires further study.

## 4. Materials and Methods

### 4.1. Study Area and Plant Material

This study was conducted at the Experimental Center of Tropical Forestry, Chinese Academy of Forestry (22°719″–22°722″ N, 106°4440″–106°4444″ E), located in Guangxi Pingxiang, China. This area experiences a subtropical monsoon climate, with long summers and abundant rainfall. The average annual temperature of Pingxiang is 19.5–21.41 °C. Rainfall mainly occurs from April to September, and the annual precipitation is approximately 1400 mm [48,49].

Seedlings of *D. odorifera*, *E. fordii*, *C. hystrix*, and *B. alnoides* were selected from nurseries in March 2014, with 90 seedlings per species. The seedlings were healthy and similar in size (approximately 20-cm tall), and were transplanted into pots filled with 5.4 L of washed river sand outdoors. From April to June 2014, three levels of irradiance, that is, 100%, 40%, and 10% of sunlight irradiance, were applied using neutral black polypropylene frames with a covering film of black polyolefin resin fine mesh. The irradiation treatment lasted for three months. Illumination was measured using an MT-4617LED-C monochromator spectroradiometer (Pro’s Kit Ltd., Shanghai, China); the average sunny midday illumination in the 100%, 40%, and 10% irradiance treatments were 78,000, 31,000, and 7800 lux, respectively.

There were three different randomized blocks per irradiance treatment, with each block consisting of 10 seedlings per species (30 seedlings per species per irradiance treatment), which were frequently moved within each block in order to avoid their position affecting the results. Each seedling was watered every day to pot water capacity and supplied with Hyponex’s nutrient solution (0.125 g N and 0.11 g P) once a week at free-access rate.

### 4.2. Determination of Gas Exchange and Fluorescence Parameters

The experiment was conducted between 09:00 a.m. and 11:00 a.m. on sunny days, on newly fully expanded leaves of seven seedlings per treatment from July to August 2014, lasting for two months. An LI-6400-40 portable photosynthesis system (LI-COR Inc., Lincoln, NE, USA) was used to determine the photosynthetic light and CO_2_ response curves. The photosynthetic response to the photosynthetic photon flux density (PPFD, µmol·m^−2^·s^−1^) was determined under a leaf chamber CO_2_ concentration of 380 μmol mol^−1^, and the net photosynthetic rate (*A*_n_, μmol·m^−2^·s^−1^), CO_2_ concentration at sub-stomatal cavities (*C*_i_, μmol mol^−1^), and stomatal conductance (*g*_s_, mol CO_2_·m^−2^·s^−1^) were measured at photon flux densities of 1500, 1200, 1000, 800, 600, 400, 200, 150, 100, 80, 50, 30, 20, 10, and 0 μmol·m^−2^·s^−1^ (see Appendix A). The PPFD-saturated net CO_2_ assimilation rate (*A*_sat_, μmol·m^−2^·s^−1^), net CO_2_ assimilation rate at PPFD of 100 umol·m^−2^·s^−1^ (*A*_100_), net CO_2_ assimilation rate at PPFD of 400 umol·m^−2^·s^−1^ (*A*_400_), dark respiration (*R*_n_, μmol·m^−2^·s^−1^), LSP (μmol·m^−2^·s^−1^), and LCP (μmol·m^−2^·s^−1^) were then measured from the light response curves. (100 and 400 umol·m^−2^·s^−1^ were in the range of the growth irradiance in the 10% and 40% light conditions, respectively). The AQY (mol·mol^−1^) was measured as the initial slope of the light response curves (PPFD ≤ 30 μmol·m^−2^·s^−1^).

The CO_2_ response curve was determined under saturated PPFD, and *C*_i_ and *g*_s_ were measured under leaf chamber CO_2_ concentrations of 380, 200, 150, 100, 80, 50, 380, 600, 800, 1000, 1200, 1500, 1800, and 2000 μmol·mol^−1^ (see Appendix A). The light- and CO_2_-saturated net CO_2_ assimilation rate (*A*_max_, μmol·m^−2^·s^−1^) was then measured from the CO_2_ response curve. The fluorescence yield (Δ*F*/*F*_m_) was measured under leaf chamber CO_2_ concentrations of 380 μmol·mol^−1^ and saturated PPFD. Meanwhile, the relative humidity of the leaf chamber was maintained at 50 ± 5%, and the leaf temperature was maintained at 25 ± 2 °C. 

### 4.3. Determination of Mesophyll Conductance, V_cmax_, and J_max_

To better measure the mesophyll conductance to CO_2_ (*g*_m_, molCO_2_ m^−2^·s^−1^), three methods were used: the variable *J* [50], exhaustive dual optimization [51], and *A*_n_–*C*_i_ curve fitting methods [52,53]. The CO_2_ concentration in chloroplasts (*C*_c_, μmol·mol^−1^) was then calculated as:(1)$$C_{C}=C_{i}-\frac{A_{\mathrm{sat}}}{g_{m}}$$

The *C*_c_ and *g*_m_ values are listed in Appendix A. The mean value of *C*_c_ calculated by the three methods was used to obtain the *A*_n__–_*C*_c_ curves, which were then used to calculate the maximum carboxylation rate (*V*_cmax_, μmol·m^−2^·s^−1^) and electron transport rates (*J*_max_, μmol·m^−2^·s^−1^) [54].

### 4.4. Determination of Additional Leaf Traits

After the determination of the gas exchange and fluorescence parameters, 20–30 leaves from each seedling used for gas exchange measurements were selected, which were healthy and similar in size. Ten to 15 of these leaves were selected, and the area of each leaf was measured using a scanner. Each leaf was then oven-dried at 80 °C for 48 h until the weight became constant, and the dry weight of each leaf was recorded. The LMA (g·m^−2^) was calculated as the ratio of the dry weight to leaf area. 

Subsequently, dries leaves were ground into powder and the leaf N per unit mass (*N*_mass_, mg·g^−1^) was determined following the micro-Kjeldahl method (UDK-139, Milano, Italy). The leaf N per unit area (*N*_area_ g·m^−2^) values were then determined as *N*_mass_ × LMA/1000, while the PNUE (μmol·mol^−1^·s^−1^) was calculated as:(2)$$\mathrm{PNUE}=\frac{A_{n}}{N_{\mathrm{area}}}\times 14$$
where PNUE_sat_ was calculated by *A*_sat_ and *N*_area_, PNUE_100_ was calculated by *A*_100_ and *N*_area_ and PNUE_400_ was calculated by *A*_400_ and *N*_area_, respectively.

The remaining 10–15 leaves from each seedling were frozen with liquid nitrogen; 0.2-g of the leaves were weighed and cut into small pieces, and then added to a volumetric flask along with 95% (*v*/*v*) alcohol to a volume of 25 mL. The flasks were then stored under darkness for 24 h. The chlorophyll content (*C*_Chl_, mmol·g^−1^) was then determined by spectrophotometry. The cell wall N concentrations (*Q*_CWmass_ mg·g^−1^) were measured following the method proposed by Onoda et al. [55], and the fraction of leaf N allocated to cell walls (*P*_CW_ g·g^−1^) was determined as *Q*_CWmass_/*N*_mass_.

### 4.5. Calculation of N Allocation in the Photosynthetic Apparatus

The N allocation proportions in Rubisco (*P*_R_, g·g^−1^), bioenergetics (*P*_B_, g·g^−1^), and light-harvesting components (*P*_L_, g·g^−1^) were calculated according to Niinemets and Tenhunen [56]:(3)$$P_{R}=\frac{V_{\mathrm{cmax}}}{6.25\times V_{\mathrm{cr}}\times \mathrm{LMA}\times N_{\mathrm{mass}}}$$
(4)$$P_{B}=\frac{J_{\max}}{8.06\times J_{\mathrm{mc}}\times \mathrm{LMA}\times N_{\mathrm{mass}}}$$
(5)$$P_{L}=\frac{C_{\mathrm{Chl}}}{C_{B}\times N_{\mathrm{mass}}}$$
where *C*_Chl_ is the chlorophyll concentration (mmol·g^−1^), *V*_cr_ is the specific activity of Rubisco (μmol CO_2_ g^−1^ Rubisco s^−1^), *J*_mc_ is the potential rate of photosynthetic electron transport (μmol electrons μmol^−1^ Cyt f s^−1^), and *C*_B_ is the ratio of leaf chlorophyll to leaf nitrogen during light-harvesting (mmol Chl (g·N)^−1^). *V*_cr_, *J*_mc_, and *C*_B_ were calculated according to Niinemets and Tenhunen [56]:(6)$$V_{\mathrm{cr}}\left(J_{\mathrm{mc}}\right)=\frac{e^{\left(c-\frac{\Delta H_{a}}{R\times T_{k}}\right)}}{1+e^{\frac{\Delta S\times T_{k}-\Delta H_{d}}{R\times T_{k}}}}$$
(7)$$[C_{B}]=1.94+\frac{12.6}{\left[\mathrm{LMA}\right]}$$
where *R* is the gas constant (8.314 J·K^−1^·mol^−1^), *T*_k_ is the leaf temperature (K), Δ*H*_a_ is the activation energy, Δ*H*_d_ is the deactivation energy, *ΔS* is the entropy term, and *c* is the scaling constant. [LMA] and [*C*_B_] are the values of LMA and *C*_B_, respectively. The values of Δ*H*_a_, Δ*H*_d_, Δ*S*, and *c* were 74,000 J·mol^−1^, 203,000 J·mol^−1^, 645 J·K^−1^·mol^−1^, and 32.9 when calculating *V*_cr_, and 24,100 J·mol^−1^, 564,150 J·mol^−1^, 1810 J·K^−1^·mol^−1^, and14.77 when calculating *J*_mc_ [56].

The leaf N allocated to the photosynthetic apparatus (*P*_P_, g·g^−1^) was calculated as *P*_R_ + *P*_B_ + *P*_L_ while the leaf N allocated to the other parts (*P*_Other_, g·g^−1^) was calculated as 1–*P*_P_–*P*_CW_. We also calculated the quantities of leaf N per unit area and the mass of N in the Rubisco, bioenergetics, light-harvesting components, photosynthetic apparatus, cell wall, and other parts (Appendix A).

### 4.6. Statistical Analysis

The differences between the four seedling species and different irradiance treatments were analyzed using one-way analysis of variance (ANOVA), and a post-hoc test (Tukey’s test) was conducted to determine if the differences were significant. The *F*-ratio in the tables is the ratio of the mean squares between groups and within groups, and *p* is the confidence interval of *F*. The significance of the linear relationships between each pair of variables was tested by Pearson’s correlation (two-tailed). All analyses were conducted using Statistical Product and Service Solutions 17.0 (version 17.0; SPSS, Chicago, IL, USA).

## 5. Conclusions

In our study, we concluded that: (1) low irradiance decreased the LMA, *N*_area_, *V*_c__max_, *J*_max_, LCP, and LSP, increased the *P*_L_ in all species; increased *A*_100_, PNUE_100_ and PNUE_400_ in N-fixing trees and decreased *A*_sat_ and *g*_s_ in non-N-fixing trees. These tree seedlings changed their leaf structure, leaf N allocation strategy, and photosynthetic physiological parameters to adapt to low light environments. (2) N-fixing plants had higher *A*_100_, *A*_400_, *N*_area_, *V*_cmax_ and *J*_max_ than non-N-fixing species under low-irradiance treatment, and were more advantageous than non-N-fixing plants in maintaining the photosynthetic rate under low-radiation conditions.

## Figures and Tables

**Figure 1 plants-10-02213-f001:**
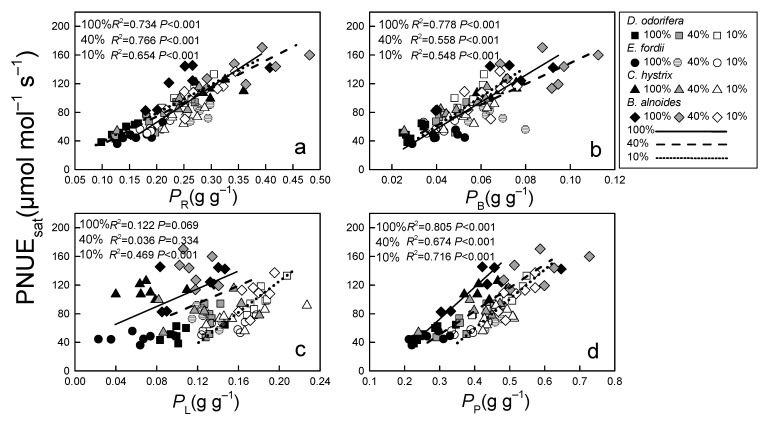
Relationship between PPFD-saturated photosynthetic N use efficiency (PNUE_sat_) and (**a**) N allocation proportion of Rubisco (*P*_R_), (**b**) bioenergetics (*P*_B_), (**c**) light-harvesting components (*P*_L_) and (**d**) photosynthetic system (*P*_P_) in *Dalbergia odorifera*, *Erythrophleum fordii, Betula alnoides*, and *Castanopsis hystrix* grown under three different irradiance treatments. The determination coefficient (*R*^2^) and *p*-value are shown.

**Figure 2 plants-10-02213-f002:**
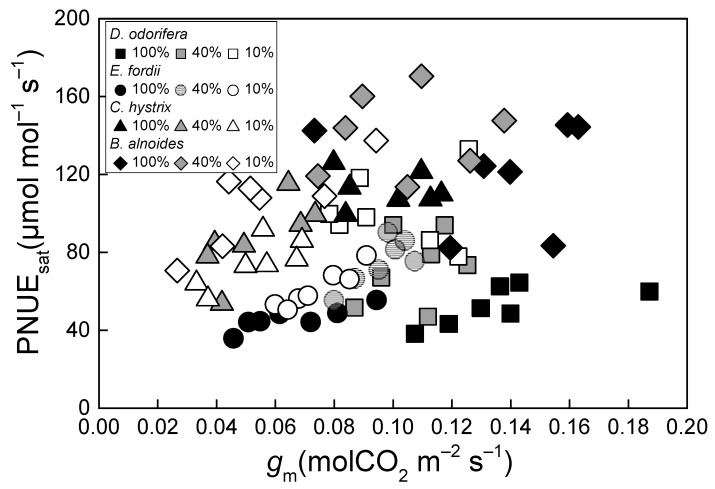
Relationship between PPFD-saturated photosynthetic N use efficiency (PNUE_sat_) and mesophyll conductance (*g*_m_) in *Dalbergia odorifera*, *Erythrophleum fordii, Betula alnoides*, and *Castanopsis hystrix* grown under three different irradiance treatments.

**Table 1 plants-10-02213-t001:** PPFD-saturated net CO_2_ assimilation rate (*A*_sat_); leaf mass per unit area (LMA); leaf nitrogen (N) content per unit of leaf area (*N*_area_); leaf N concentration (*N*_mass_) and PPFD-saturated photosynthetic N use efficiency (PNUE_sat_) in *Dalbergia odorifera*, *Erythrophleum fordii, Betula alnoides*, and *Castanopsis hystrix* grown under three different irradiance treatments. Data are means of seven plants per treatment ±SE. Lower case letters indicate significant difference at 0.05 levels among the irradiance treatments, whereas capital letters indicate significant difference at 0.05 levels among species under same irradiance treatment. *F*-ratios with statistically significant values denoted by * *p* < 0.05, ** *p* < 0.01, *** *p* < 0.001 among irradiance treatment.

Tree Species	Irradiance Treatment	*A*_sat_ (μmol·m^−2^·s^−1^)	*N*_area_ (g·m^−2^)	*N*_mass_ (mg·g^−1^)	LMA (g·m^−2^)	PNUE_sat_ (μmol·mol^−1^·s^−1^)
*Dalbergia odorifera*	100%	8.04 ± 0.46 ^aA^	2.19 ± 0.13 ^aA^	31.7 ± 0.76 ^aA^	69.0 ± 3.90 ^aB^	52.6 ± 3.78 ^bB^
	40%	8.30 ± 0.76 ^aA^	1.62 ± 0.04 ^bA^	31.2± 0.65 ^aA^	51.8 ± 0.65 ^bB^	72.3 ± 7.03 ^bB^
	10%	6.88 ± 0.30 ^aA^	0.97 ± 0.04 ^cB^	33.0 ± 1.11 ^aA^	29.3 ± 0.67 ^cC^	101.0 ± 7.12 ^aA^
	*F*	1.967	54.700 ***	1.196	73.752 ***	15.533 ***
*Erythrophleum fordii*	100%	6.60 ± 0.50 ^bB^	2.01 ± 0.12 ^aA^	28.1 ± 1.49 ^bB^	71.4 ± 0.89 ^aB^	45.9 ± 2.24 ^cB^
	40%	9.34 ± 0.49 ^aA^	1.75 ± 0.03 ^bA^	33.0 ± 0.46 ^bA^	53.1 ± 0.99 ^bB^	75.0 ± 4.56 ^aB^
	10%	6.87 ± 0.50 ^bA^	1.56 ± 0.04 ^bA^	35.3 ± 0.88 ^aA^	44.3 ± 1.47 ^cB^	61.6 ± 3.72 ^bB^
	*F*	9.042 **	9.223 **	12.658 ***	145.227 ***	15.877 ***
*Castanopsis hystrix*	100%	8.16 ± 0.18 ^aA^	1.02 ± 0.06 ^aB^	10.2 ± 1.80 ^bD^	100.1 ± 2.60 ^aA^	112.0 ± 4.62 ^aA^
	40%	4.57 ± 0.23 ^bB^	0.75 ± 0.05 ^bB^	9.6 ± 0.50 ^bC^	78.8 ± 1.11 ^bA^	87.0± 7.26 ^bB^
	10%	4.18 ± 0.25 ^bB^	0.79 ± 0.03 ^bC^	13.7 ± 0.49 ^aC^	57.9 ± 1.29 ^cA^	74.4 ± 4.59 ^bB^
	*F*	95.630 ***	20.060 ***	28.220 ***	138.877 ***	12.868 ***
*Betula alnoides*	100%	8.55 ± 0.60 ^aA^	1.03 ± 0.09 ^aB^	15.4 ± 1.04 ^bC^	67.6 ± 5.45 ^aB^	120.5 ± 5.18 ^abA^
	40%	7.42 ± 0.30 ^aA^	0.75 ± 0.04 ^bB^	15.4 ± 0.45 ^bB^	49.1 ± 3.36 ^bB^	140.3 ± 8.02 ^aA^
	10%	4.26 ± 0.52 ^bB^	0.56 ± 0.04 ^bD^	19.0 ± 0.62 ^aB^	29.6 ± 2.14 ^cC^	105.3 ± 8.33 ^bA^
	*F*	20.458 ***	13.371 ***	7.790 **	23.833 ***	3.815 *

**Table 2 plants-10-02213-t002:** Stomatal conductance (*g*_s_), mesophyll conductance (*g*_m_), intercellular CO_2_ concentration (*C*_i_), CO_2_ concentration at carboxylation site (*C*_c_) and CO_2_ drawdown from the intercellular concentration to the carboxylation site concentration (*C*_i_-*C*_c_) measured in PPFD-saturated conditions in *Dalbergia odorifera*, *Erythrophleum fordii, Betula alnoides*, and *Castanopsis hystrix* grown under three different irradiance treatments. Data are means of seven plants per treatment ±SE. Lower case letters indicate significant difference at 0.05 levels among the irradiance treatments, whereas capital letters indicate significant difference at 0.05 levels among species under same irradiance treatment. *F*-ratios with statistically significant values denoted by * *p* < 0.05, ** *p* < 0.01, *** *p* < 0.001 among irradiance treatment.

Tree Species	Irradiance Treatment	*g*_s_ (molCO_2_·m^−2^·s^−1^)	*g*_m_ (molCO_2_·m^−2^·s^−1^)	*C*_i_ (μmol·mol^−1^)	*C*_c_ (μmol·mol^−1^)	*C*_i_-*C*_c_ (μmol·mol^−1^)
*Dalbergia odorifera*	100%	0.067 ± 0.004 ^bBC^	0.137 ± 0.010 ^aA^	251.5 ± 6.44 ^bBC^	190.8 ± 6.92 ^bB^	60.8 ± 2.21 ^aC^
	40%	0.091 ± 0.009 ^aA^	0.107 ± 0.005 ^bA^	288.5 ± 3.93 ^aA^	210.0 ± 8.82 ^bA^	78.6 ± 7.50 ^aAB^
	10%	0.089 ± 0.003 ^aA^	0.100 ± 0.007 ^bA^	302.6 ± 1.94 ^aA^	231.3 ± 6.20 ^aA^	71.2 ± 5.27 ^aB^
	*F*	6.562 **	6.823 **	34.333 ***	7.536 **	2.698
*Erythrophleum fordii*	100%	0.046 ± 0.002 ^bC^	0.066 ± 0.007 ^bC^	235.6 ± 6.19 ^bC^	132.6 ± 6.90 ^bD^	103.0 ± 4.83 ^aA^
	40%	0.075 ± 0.005 ^aA^	0.096 ± 0.004 ^aA^	254.1 ± 3.81 ^abB^	156.8 ± 4.09 ^aB^	97.3 ± 2.37 ^aA^
	10%	0.060 ± 0.005 ^abB^	0.074 ± 0.004 ^bB^	264.4 ± 4.18 ^aB^	171.7 ± 4.45 ^aB^	92.7 ± 1.56 ^aA^
	*F*	11.744 **	9.789 **	7.982 **	13.855 ***	2.553
*Castanopsis hystrix*	100%	0.074 ± 0.004 ^aB^	0.099 ± 0.006 ^aB^	256.8 ± 5.24 ^bB^	168.0 ± 6.04 ^bC^	88.8 ± 6.26 ^aB^
	40%	0.039 ± 0.004 ^bB^	0.053 ± 0.006 ^bB^	280.1 ± 3.48 ^aA^	196.4 ± 5.84 ^aA^	83.7 ± 4.14 ^aAB^
	10%	0.036 ± 0.001 ^bC^	0.053 ± 0.005 ^bB^	268.0 ± 5.65 ^abB^	186.5 ± 6.27 ^abB^	81.5 ± 5.03 ^aAB^
	*F*	38.06 ***	22.353 ***	5.711 *	5.691 *	0.511
*Betula alnoides*	100%	0.100 ± 0.013 ^aA^	0.134 ± 0.012 ^aA^	292.9 ± 5.94 ^aA^	226.4 ± 9.57 ^aA^	66.5 ± 4.64 ^aC^
	40%	0.095 ± 0.011 ^aA^	0.104 ± 0.009 ^aA^	297.1 ± 7.04 ^aA^	222.9 ± 12.19 ^aA^	74.2 ± 5.69 ^aB^
	10%	0.063 ± 0.005 ^bB^	0.056 ± 0.009 ^bB^	312.3 ± 4.73 ^aA^	232.1 ± 9.07 ^aA^	80.3 ± 6.14 ^aAB^
	*F*	4.195 *	16.261 ***	2.93	0.198	1.568

**Table 3 plants-10-02213-t003:** Maximum carboxylation rate (*V*_cmax_) and maximum electron transport rate (*J*_max_) measured in PPFD-saturated conditions in *Dalbergia odorifera*, *Erythrophleum fordii, Betula alnoides*, and *Castanopsis hystrix* grown under three different irradiance treatments obtained by fitting the Farquhar et al. (1980) model of leaf photosynthesis to the individual *A*_n_-*C*_c_ response curves. Data are means of seven plants per treatment ±SE. Lower case letters indicate significant difference at 0.05 levels among the irradiance treatments, whereas capital letters indicate significant difference at 0.05 levels among species under same irradiance treatment. *F*-ratios with statistically significant values denoted by * *p* < 0.05, *** *p* < 0.001 among irradiance treatment.

Tree Species	Irradiance Treatment	*V*_cmax_ (μmol·m^−2^·s^−1^)	*J*_max_ (μmol·m^−2^·s^−1^)
*Dalbergia odorifera*	100%	78.1 ± 4.59 ^bB^	100.7 ± 5.80 ^aBC^
	40%	95.2 ± 8.01 ^aB^	118.5 ± 7.39 ^aB^
	10%	68.6 ± 3.96 ^cB^	79.1 ± 2.76 ^bB^
	*F*	5.405 *	12.154 ***
*Erythrophleum fordii*	100%	99.8 ± 9.37 ^bA^	128.8 ± 11.20 ^bAB^
	40%	141.4 ± 5.24 ^aA^	168.9 ± 3.36 ^aA^
	10%	80.1 ± 4.07 ^cA^	99.8 ± 3.83 ^cA^
	*F*	22.233 ***	23.930 ***
*Castanopsis hystrix*	100%	82.8 ± 4.47 ^aB^	109.3 ± 3.40 ^aABC^
	40%	46.5 ± 2.51 ^bC^	57.6 ± 4.49 ^bC^
	10%	47.7 ± 2.92 ^bC^	66.5 ± 3.80 ^bC^
	*F*	75.031 ***	49.677 ***
*Betula alnoides*	100%	73.0 ± 3.51 ^aB^	98.4 ± 5.37 ^aBC^
	40%	82.6 ± 5.46 ^aB^	97.8 ± 5.39 ^aB^
	10%	41.6 ± 4.80 ^bC^	56.0 ± 4.59 ^bC^
	*F*	25.05 ***	22.465 ***

**Table 4 plants-10-02213-t004:** Nitrogen allocation proportion of Rubisco (*P*_R_), bioenergetics (*P*_B_), light-harvesting components (*P*_L_), photosynthetic system (*P*_P_), cell wall (*P*_CW_) and other parts (*P*_Other_) in *Dalbergia odorifera*, *Erythrophleum fordii, Betula alnoides*, and *Castanopsis hystrix* grown under three different irradiance treatments. Data are means of seven plants per treatment ±SE. Lower case letters indicate significant difference at 0.05 levels among the irradiance treatments, whereas capital letters indicate significant difference at 0.05 levels among species under same irradiance treatment. *F*-ratios with statistically significant values denoted by * *p* < 0.05, ** *p* < 0.01, *** *p* < 0.001 among irradiance treatment.

Tree Species	Irradiance Treatment	*P*_R_ (g·g^−1^)	*P*_B_ (g·g^−1^)	*P*_L_ (g·g^−1^)	*P*_P_ (g·g^−1^)	*P*_CW_ (g·g^−1^)	*P*_Other_ (g·g^−1^)
*Dalbergia odorifera*	100%	0.135 ± 0.009 ^bB^	0.030 ± 0.002 ^bB^	0.105 ± 0.008 ^cA^	0.269 ± 0.016 ^cB^	0.068 ± 0.004 ^aC^	0.663 ± 0.015 ^aA^
	40%	0.201 ± 0.018 ^aA^	0.047 ± 0.003 ^aC^	0.132 ± 0.002 ^bA^	0.381 ± 0.021 ^bB^	0.067 ± 0.006 ^aC^	0.552 ± 0.017 ^bA^
	10%	0.242 ± 0.016 ^aAB^	0.054 ± 0.003 ^aA^	0.183 ± 0.005 ^aA^	0.479 ± 0.021 ^aA^	0.061 ± 0.004 ^aC^	0.461 ± 0.023 ^cB^
	*F*	14.001 ***	27.585 ***	54.347 ***	29.423 ***	0.632	29.390 ***
*Erythrophleum fordii*	100%	0.164 ± 0.010 ^cB^	0.043 ± 0.003 ^bB^	0.060 ± 0.009 ^cB^	0.266 ± 0.018 ^cB^	0.052 ± 0.002 ^aC^	0.683 ± 0.019 ^aA^
	40%	0.268 ± 0.011 ^aB^	0.065 ± 0.003 ^aB^	0.129 ± 0.004 ^bA^	0.462 ± 0.007 ^aB^	0.038 ± 0.001 ^bC^	0.500 ± 0.007 ^cA^
	10%	0.203 ± 0.011 ^bB^	0.041 ± 0.002 ^bB^	0.150 ± 0.008 ^aB^	0.394 ± 0.016 ^bB^	0.039 ± 0.002 ^bC^	0.568 ± 0.015 ^bA^
	*F*	24.021 ***	25.215 ***	38.638 ***	47.577 ***	24.303 ***	40.909 ***
*Castanopsis hystrix*	100%	0.302 ± 0.012 ^aA^	0.068 ± 0.003 ^aA^	0.072 ± 0.008 ^bB^	0.441 ± 0.018 ^aA^	0.267 ± 0.010 ^bA^	0.292 ± 0.019 ^aB^
	40%	0.231 ± 0.018 ^bB^	0.049 ± 0.005 ^bC^	0.130 ± 0.014 ^aA^	0.411 ± 0.032 ^aB^	0.443 ± 0.022 ^aA^	0.146 ± 0.023 ^cB^
	10%	0.247 ± 0.010 ^bAB^	0.054 ± 0.003 ^bA^	0.164 ± 0.013 ^aAB^	0.466 ± 0.015 ^aA^	0.342 ± 0.028 ^bA^	0.192 ± 0.031 ^bD^
	*F*	7.010 **	6.229 **	15.540 ***	1.475	17.205 ***	4.023 *
*Betula alnoides*	100%	0.256 ± 0.028 ^bA^	0.066 ± 0.007 ^bA^	0.116 ± 0.011 ^bA^	0.439 ± 0.042 ^cA^	0.221 ± 0.011 ^aB^	0.340 ± 0.042 ^aB^
	40%	0.369 ± 0.026 ^aB^	0.089 ± 0.006 ^aA^	0.119 ± 0.005 ^bA^	0.577 ± 0.033 ^aA^	0.197 ± 0.011 ^aB^	0.227 ± 0.031 ^aB^
	10%	0.281 ± 0.017 ^bA^	0.063 ± 0.003 ^bA^	0.176 ± 0.005 ^aAB^	0.521 ± 0.016 ^bA^	0.150 ± 0.010 ^bB^	0.329 ± 0.021 ^aC^
	*F*	5.979 **	6.189 **	20.386 ***	4.641 *	10.826 **	3.774 *

**Table 5 plants-10-02213-t005:** Apparent quantum yield (AQY), dark respiration (*R*_n_), light compensation point (LCP) and light saturation point (LSP) in *Dalbergia odorifera*, *Erythrophleum fordii*, *Betula alnoides*, and *Castanopsis hystrix* grown under three different irradiance treatments. Data are means of seven plants per treatment ±SE. Lower case letters indicate significant difference at 0.05 levels among the irradiance treatments, whereas capital letters indicate significant difference at 0.05 levels among species under same irradiance treatment. *F*-ratios with statistically significant values denoted by * *p* < 0.05, ** *p* < 0.01, *** *p* < 0.001 among irradiance treatment.

Tree Species	Irradiance Treatment	AQY (mol·mol^−1^)	*R*_n_ (μmol·m^−2^·s^−1^)	LCP (μmol·m^−2^·s^−1^)	LSP (μmol·m^−2^·s^−1^)
*Dalbergia odorifera*	100%	0.052 ± 0.004 ^aA^	0.909 ± 0.050 ^aBC^	22.1 ± 1.68 ^aA^	822.9 ± 27.5 ^aA^
	40%	0.059 ± 0.002 ^aA^	0.845 ± 0.050 ^aA^	14.4 ± 0.73 ^bB^	724.3 ± 37.0 ^aBA^
	10%	0.058 ± 0.002 ^aA^	0.760 ± 0.038 ^aB^	7.4 ± 0.43 ^cB^	684.3 ± 23.1 ^bA^
	*F*	2.300	2.587	46.127 ***	5.378 *
*Erythrophleum fordii*	100%	0.047 ± 0.003 ^bA^	1.129 ± 0.051 ^aA^	13.9 ± 0.81 ^aB^	637.1 ± 29.6 ^aB^
	40%	0.062 ± 0.002 ^aA^	0.873 ± 0.050 ^bA^	13.1 ± 1.10 ^aB^	633.6 ± 17.1 ^aA^
	10%	0.059 ± 0.001 ^aA^	0.936 ± 0.030 ^bAB^	7.5 ± 0.79 ^bB^	522.1 ± 17.2 ^bB^
	*F*	15.924 ***	8.811 **	14.464 ***	6.569 **
*Castanopsis hystrix*	100%	0.047 ± 0.003 ^aA^	1.005 ± 0.067 ^aAB^	14.0 ± 1.21 ^aB^	632.9 ± 23.4 ^aB^
	40%	0.054 ± 0.002 ^aA^	0.988 ± 0.040 ^aA^	7.3 ± 1.00 ^bC^	307.1 ± 26.8 ^bC^
	10%	0.054 ± 0.003 ^aA^	1.048 ± 0.088 ^aA^	7.4 ± 1.21 ^bB^	262.1 ± 27.7 ^bC^
	*F*	2.737	0.206	11.226 **	60.359 ***
*Betula alnoides*	100%	0.049 ± 0.001 ^aA^	0.889 ± 0.039 ^aBC^	20.8 ± 0.93 ^aA^	886.4 ± 43.5 ^aA^
	40%	0.055 ± 0.003 ^aA^	0.844 ± 0.055 ^aA^	18.6 ± 2.49 ^aA^	519.3 ± 27.9 ^bB^
	10%	0.051 ± 0.003 ^aA^	0.834 ± 0.048 ^aAB^	10.5 ± 0.94 ^bA^	268.6 ± 30.5 ^cC^
	*F*	1.415	0.384	10.989 **	80.343 ***

**Table 6 plants-10-02213-t006:** Net CO_2_ assimilation rate at PPFD of 100 umol·m^−2^·s^−1^ (*A*_100_); net CO_2_ assimilation rate at PPFD of 400 umol·m^−2^·s^−1^ (*A*_400_); photosynthetic N use efficiency at PPFD of 100 umol·m^−2^·s^−1^ (PNUE_100_); photosynthetic N use efficiency at PPFD of 400 umol·m^−2^·s^−1^ (PNUE_400_) in *Dalbergia odorifera*, *Erythrophleum fordii*, *Betula alnoides*, and *Castanopsis hystrix* grown under three different irradiance treatments. Data are means of seven plants per treatment ± SE. Lower case letters indicate significant difference at 0.05 levels among the irradiance treatments, whereas capital letters indicate significant difference at 0.05 levels among species under same irradiance treatment. F-ratios with statistically significant values denoted by * *p* < 0.05, ** *p* < 0.01, *** *p* < 0.001 among irradiance treatment.

Tree Species	Irradiance Treatment	*A*_100_ (μmol·m^−2^·s^−1^)	*A*_400_ (μmol·m^−2^·s^−1^)	PNUE_100_ (μmol·mol^−1^·s^−1^)	PNUE_400_ (μmol·mol^−1^·s^−1^)
*Dalbergia odorifera*	100%	2.78 ± 0.41 ^bA^	7.38 ± 0.45 ^abAB^	20.7 ± 1.36 ^cB^	48.2 ± 3.65 ^cB^
	40%	4.06 ± 0.08 ^aA^	8.02 ± 0.30 ^aA^	35.2 ± 0.66 ^bB^	69.8 ± 3.35 ^bB^
	10%	4.03 ± 0.17 ^aA^	6.65 ± 0.36 ^bA^	59.2 ± 4.12 ^aAB^	97.4 ± 7.32 ^aA^
	*F*	7.114 *	6.206 *	58.81 ***	23.261 ***
*Erythrophleum fordii*	100%	3.42 ± 0.16 ^bA^	6.07 ± 0.24 ^bB^	24.2 ± 1.38 ^bB^	42.7 ± 1.48 ^cB^
	40%	4.04 ± 0.08 ^aA^	8.53 ± 0.22 ^aA^	32.3 ± 0.72 ^aB^	68.4 ± 2.42 ^aB^
	10%	3.98 ± 0.18 ^aA^	6.31 ± 0.43 ^bA^	35.8 ± 1.29 ^aC^	56.6 ± 3.11 ^bB^
	*F*	5.426 *	16.016 ***	26.501 ***	22.155 ***
*Castanopsis hystrix*	100%	3.57 ± 0.12 ^aA^	7.67 ± 0.42 ^aAB^	49.1 ± 1.83 ^aA^	105.4 ± 6.18 ^aA^
	40%	3.00 ± 0.15 ^bB^	3.98 ± 0.29 ^bB^	57.3 ± 4.57 ^aA^	76.2 ± 8.00 ^bB^
	10%	2.92 ± 0.19 ^bB^	3.68 ± 0.31 ^bB^	51.8 ± 3.22 ^aBC^	65.3 ± 4.99 ^bB^
	*F*	5.107 *	41.202 ***	1.506	10.123 **
*Betula alnoides*	100%	3.27 ± 0.15 ^aA^	7.95 ± 0.52 ^aA^	46.4 ± 4.09 ^bA^	113.7 ± 12.66 ^aA^
	40%	3.52 ± 0.28 ^aAB^	7.16 ± 0.68 ^aA^	67.3 ± 7.62 ^abA^	135.9 ± 14.91 ^aA^
	10%	2.92 ± 0.32 ^aB^	3.91 ± 0.66 ^bB^	72.9 ± 6.33 ^aA^	95.6 ± 11.85 ^aA^
	*F*	1.358	12.425 ***	5.074 *	2.852

## Data Availability

The data presented in this study are available within this article and Appendix A.

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
