# Peer review of "The Effect of Low Irradiance on Leaf Nitrogen Allocation and Mesophyll Conductance to CO2 in Seedlings of Four Tree Species in Subtropical China"

_plants, 2021, doi:10.3390/plants10102213_

Round 1
Reviewer 1 Report
The manuscript by Tang et al. reports on the effect of high nitrogen content of nitrogen-fixing plants on photosynthetic rate under low light radiation conditions. The authors analyzed the leaf morphology (e.g. leaf area, thickness), physiological (e.g. photosynthetic rate, stomatal conductance, mesophyll conductance, photosynthetic nitrogen use efficiency) and leaf N allocation characteristics of four species (two non- N-fixing tree species, and two N-fixing tree species) under three irradiance levels. The results demonstrate essential differences in changes of leaf traits of four species under three irradiance conditions. Nitrogen-fixing plants maintain photosynthetic rates by increasing Ci and Cc under 10% light radiation conditions. Under the 10% irradiation treatment, the photosynthetic nitrogen use efficiency of the two nitrogen-fixing trees was higher than that of the non-nitrogen-fixing plants because the proportion of nitrogen in the photosystem increased. Without doubt, the topic is of urgent current interest. However, I think the paper still needs further improvement before publication.
General comments
1) The “introduction” section is weak and non-specific, and the statement of objectives/questions wasn’t logical and clear. Please re-write it and explain better why your study could contribute to current knowledge.
2) The critical results (the effects of 40% irradiance on plants) were not presented in “Results” and “Discussion” sections.
3) The results were not enough to make a convincing conclusion. The conclusions appear overstated and without sufficient results or rationale to explain how they were arrived at.
4) I strongly recommend the authors to use a professional English editing service.
Specific comments
1) L35-37: Please clarify why it is important to study low light radiation. For example, low radiation conditions in the understory canopy of subtropical forests affect the growth rate of seedlings.
2) L52-54: As far as I know, Ci and Cc are regulated by gs, gm and Vcmax. Please add references to justify the reasoning.
3) L65-66: Please clarify the possible advantages of nitrogen-fixing plants in dealing with low light radiation.
4) L70-71: Please clarify the reasons for choosing the four species in the manuscript. Are they the dominant species in the study area?
5) L73-75: These two sentences have nothing to do with your research, please delete.
6) L75-77: I would suggest rewrite this sentence as "The aim of this work was to address the following objectives: (1) to determine the effects of low irradiance on the leaf structure, leaf nitrogen (N) allocation strategy and photosynthetic physiological parameters (e.g. gs, gm and photosynthetic rate); (2) to evaluated whether nitrogen-fixing plants are more advantageous than non- fixing plants in maintaining photosynthetic rate under low-radiation conditions.”. I think it can deliver a clear message to the reader.
7) L35-77: Please re-organize and re-write the section “Introduction” to clarify the knowledge gaps in current research
8) L251-253: What is the rationale for selection 40% and 10% radiation? Is this the radiation intensity of the understory canopy?
9) L267-271: Please add specific information (e.g. CO2 concentration and light gradients) about the CO2 and light response curve?
10) Figure 3: Please improve figure quality.
11) Please add the critical results (the effects of 40% irradiance on plants) w in “Results” section.
12) The authors should bring to the manuscript a more reliable discussion in terms of how such a study can contribute to the advance in the literature regarding the importance of low irradiance on photosynthetic rates and the advantage of nitrogen-fixing species in maintain photosynthetic rate under low irradiance.
Author Response
Dear reviewer
Re: Manuscript ID: plants-1377753
Please find attached a revised version of our manuscript “The effect of low irradiance on leaf nitrogen allocation and mesophyll conductance to CO2 in seedlings of four tree species in subtropical China”, which we would like to resubmit for publication as a research paper in Plants.
Your comments were highly insightful and enabled us to greatly improve the quality of our manuscript. In the following pages are our point-by-point responses to each of the comments of the reviewers as well as your own comments.
Revisions in the text are shown using yellow highlight for additions, and strikethrough font [example] for deletions. We hope that the revisions in the manuscript and our accompanying responses will be sufficient to make our manuscript suitable for publication in Plants.
We shall look forward to hearing from you at your earliest convenience.
Thank you and best regards.
Yours sincerely,
Jingchao Tang
Ph: (+86)18866482400
E–mail: tjc@qut.edu.cn
Responses to the comments of Reviewer #1
1.General comments
Question 1: The “introduction” section is weak and non-specific, and the statement of objectives/questions wasn’t logical and clear. Please re-write it and explain better why your study could contribute to current knowledge.
Response1: We have rewritten the “introduction” section, rewritten the statement of objectives/questions, and explained the contribute to current knowledge of our study. Please see “introduction” section in our manuscript of revised version.
Question 2: The critical results (the effects of 40% irradiance on plants) were not presented in “Results” and “Discussion” sections.
Response: We have rewritten the “Results” and “Discussion” sections, and added “the effects of 40% irradiance on plants” in these sections. Please see “Results” and “Discussion” sections in our manuscript of revised version.
Question 3: The results were not enough to make a convincing conclusion. The conclusions appear overstated and without sufficient results or rationale to explain how they were arrived at.
Response: We have rewritten the “Results” and “Conclusion” sections, please see these sections in our manuscript of revised version.
Question 4: I strongly recommend the authors to use a professional English editing service.
Response: Our manuscript was edited by professional editors at Editage to improve English, please see attachment for proof materials
- Specific comments
Question 1:. L35-37: Please clarify why it is important to study low light radiation. For example, low radiation conditions in the understory canopy of subtropical forests affect the growth rate of seedlings.
Response: We added the importance of study low light radiation in seedlings
[Line 36 to 39 in revised version]
Radiation is a source of energy for plants. Through photosynthesis, green plants use light to synthesize carbohydrates from water and CO2, which are necessary for maintaining growth and development. Low radiation conditions in the understory canopy of subtropical forests affect the survival and growth of forest tree seedlings [1].
Question 2:. L52-54: As far as I know, Ci and Cc are regulated by gs, gm and Vcmax. Please add references to justify the reasoning.
Response: You are right, the express in this place was wrong. We have modified this sentence and added references.
[Line 50 to 58 in revised version]
Under low irradiance, the leaf thickness may decrease and the leaf area may increase, resulting in a lower leaf mass per unit area (LMA) [1,4,14], which increases the area receiving light [8]. Low irradiance could also result in a reduction in the surface area of mesophyll cells per unit leaf area, as well as a smaller area of mesophyll cells through which CO2 can diffuse into the chlorophyll [15–16]. These changes subsequently affect the mesophyll conductance to CO2 (gm) and, in turn, affect the CO2 concentration in chloroplasts (Cc) [17-18]. Low irradiance decreased gm [19–20] or did not significantly affect gm in different species [21–22]. Therefore, changes in gm in different species should be further studied.
Question 3:. L65-66: Please clarify the possible advantages of nitrogen-fixing plants in dealing with low light radiation.
Response: We clarified the possible advantages of nitrogen-fixing plants in dealing with low light radiation.
[Line 66 to 71 in revised version]
However, few studies have been conducted on whether low-irradiance treatment can affect the PNUE of N-fixing trees and the relevant internal control mechanism of leaf N allocation and gm. We suspect that N-fixing species with sufficient N in their leaves could increase their PR, PB, and PL to increase the PNUE under low-irradiance treatment, and maintain their photosynthetic capacity and growth better than non-N-fixing species under low-irradiance environments.
Question 4:. L70-71: Please clarify the reasons for choosing the four species in the manuscript. Are they the dominant species in the study area?
Response: We clarify the reasons for choosing the four species in the manuscript.
[Line 75 to 80 in revised version]
In this study, we exposed Dalbergia odorifera and Erythrophleum fordii (N-fixing trees), and Castanopsis hystrix and Betula alnoides (non-N-fixing trees) seedlings to three levels of irradiance (100%, 40%, and 10% of sunlight irradiance) and estimated their photosynthesis, PNUE, leaf N allocation, and gm values. These species are locally vital broad-leaved trees with high economic values that are commonly used to change Pinus massoniana and Cunninghamia lanceolata pure forests into mixed broadleaf-conifer forests or mixed broad-leaved forests. This requires the selected species to be planted in forest gaps, mixed with other species, or directly on bare ground; therefore, their tolerance under low light conditions (e.g., in the understory canopy) will affect their survival and growth.
Question 5:. L73-75: These two sentences have nothing to do with your research, please delete.
Response: We deleted these two sentences.
[Line 72 to 75 in revised version]
In this study, we exposed Dalbergia odorifera and Erythrophleum fordii (N-fixing trees), and Castanopsis hystrix and Betula alnoides (non-N-fixing trees) seedlings to three levels of irradiance (100%, 40%, and 10% of sunlight irradiance) and estimated their photosynthesis, PNUE, leaf N allocation, and gm values. We previously studied the interspecific differences between these trees [28], and the effects of soil N deficiency on these trees [29].
Question 6:. L75-77: I would suggest rewrite this sentence as "The aim of this work was to address the following objectives: (1) to determine the effects of low irradiance on the leaf structure, leaf nitrogen (N) allocation strategy and photosynthetic physiological parameters (e.g. gs, gm and photosynthetic rate); (2) to evaluated whether nitrogen-fixing plants are more advantageous than non- fixing plants in maintaining photosynthetic rate under low-radiation conditions.”. I think it can deliver a clear message to the reader.
Response: Thank you very much for your guidance, we have rewritten this sentence.
[Line 83 to 86 in revised version]
The aim of this study was to (1) determine the effects of low irradiance on leaf structure, leaf N allocation strategy, and photosynthetic physiological parameters (e.g., gs, gm, and photosynthetic rate) and (2) evaluate whether N-fixing plants are better able to maintain their photosynthetic rate under low-radiation conditions than non-fixing plants.
Question 7:. L35-77: Please re-organize and re-write the section “Introduction” to clarify the knowledge gaps in current research
Response: We re-organize and re-write the section “Introduction”, and clarified the knowledge gaps in current research. Please see “introduction” section in our manuscript of revised version.
Question 8:. L251-253: What is the rationale for selection 40% and 10% radiation? Is this the radiation intensity of the understory canopy?
Response: 10% and 40% full light conditions are common light conditions for forest gap and mixed planting, while 100% light conditions are light conditions for direct planting in bare land. We added this explanation in our manuscript.
[Line 78 to 82 in revised version]
This requires the selected species to be planted in forest gaps, mixed with other species, or directly on bare ground; therefore, their tolerance under low light conditions (e.g., in the understory canopy) will affect their survival and growth. Full light conditions of 10% and 40% are common in forest gaps and mixed planting conditions, while 100% light conditions are typical for direct planting on bare land.
Question 9:. L267-271: Please add specific information (e.g. CO2 concentration and light gradients) about the CO2 and light response curve?
Response: We added CO2 concentration and light gradients about the CO2 and light response curve in our manuscript.
[Line 337 to 356 in revised version]
The experiment was conducted between 09:00 and 11:00 h on sunny days, on newly fully expanded leaves of seven seedlings per treatment from July to August 2014, lasting for two months. An LI-6400-40 portable photosynthesis system (LI-COR, Lincoln, Nebraska, USA) was used to determine the photosynthetic light and CO2 response curves. The photosynthetic response to the photosynthetic photon flux density (PPFD, µmol m-2 s-1) was determined under a leaf chamber CO2 concentration of 380 μmol mol–1, and the net photosynthetic rate (An, μmol m–2 s–1), CO2 concentration at sub-stomatal cavities (Ci, μmol mol–1), and stomatal conductance (gs, mol CO2 m–2 s–1) were measured at photon flux densities of 1500, 1200, 1000, 800, 600, 400, 200, 150, 100, 80, 50, 30, 20, 10, and 0 μmol m–2 s–1 (see Fig. S1). The PPFD-saturated net CO2 assimilation rate (Asat, μmol m–2 s–1), dark respiration (Rn, μmol m–2 s–1), LSP (μmol m–2 s–1), and LCP (μmol m–2 s–1) were then measured from the light response curves. The AQY (mol mol–1) was measured as the initial slope of the light response curves (PPFD ≤ 200 μmol m–2 s–1).
The CO2 response curve was determined under saturated PPFD, and Ci and gs were measured under leaf chamber CO2 concentrations of 380, 200, 150, 100, 80, 50, 380, 600, 800, 1000, 1200, 1500, 1800, and 2000 μmol mol–1 (see Fig. S2). The light- and CO2-saturated net CO2 assimilation rate (Amax, μmol m–2 s–1) was then measured from the CO2 response curve. The fluorescence yield (⊿F/Fm) was measured under leaf chamber CO2 concentrations of 380 μmol mol–1 and saturated PPFD. Meanwhile, the relative humidity of the leaf chamber was maintained at 50 ± 5%, and the leaf temperature was maintained at 25 ± 2 °C.
Question 10:. Figure 3: Please improve figure quality.
Response: We have redrawn Figure 3
[Line 305 in revised version]
Question 11:. Please add the critical results (the effects of 40% irradiance on plants) w in “Results” section.
Response: We added the effects of 40% irradiance on plants in “Results” section, and have rewritten “Results” section. Please see “Results” section in our manuscript of revised version.
Question 12:. The authors should bring to the manuscript a more reliable discussion in terms of how such a study can contribute to the advance in the literature regarding the importance of low irradiance on photosynthetic rates and the advantage of nitrogen-fixing species in maintain photosynthetic rate under low irradiance.
Response: We added one passage about the importance of low irradiance on photosynthetic rates and the advantage of nitrogen-fixing species in maintain photosynthetic rate under low irradiance in “Discussion” section.
[Line 261 to 276 in revised version]
As these species are commonly used to plant in gaps or mixed with other species, their tolerance to low light conditions will affect the growth effect after planting. All four species decreased the LMA to increase the area receiving light (Table 1) [1,4,8,14], increased PL to increase their light-harvesting capacity and sustain photosynthesis (Table 4) [3–4], and decreased the LCP and LSP to increase the ability to use low light (Table 5) under low light conditions. Meanwhile, N-fixing plants exhibited some other adaptations to low light conditions, such as increased Ci and Cc, maintained Asat under the 10% and 40% irradiance treatments (Tables 1 and 2), and increased PNUE under the 10% and 40% irradiance treatments (Table 1). N-fixing plants also had higher Asat, Narea, Vcmax, and Jmax than non-N-fixing species under the 10% irradiance treatment (Table 1,3). AQY refers to the ability to use low light [43] E. Fordii exhibited improved AQY under the 10% and 40% irradiance treatments, and the AQY values under the 10% irradiance treatment in the order of: E. fordii > D. odorifera > C. hystrix > B. Alnoides (Table 5). E. fordii also reduced Rn under the 10% and 40% irradiance treatments to reduce respiratory expenditure (Table 5). In conclusion, these results suggest that the adaptability of N-fixing species to low light environments is better than that of non-N-fixing species.

Reviewer 2 Report
The present research studied the response of leaf physiological traits of 4 tree species to the 3 different light intensities. Relatively large number of leaf physiological traits were studied but the interpretation of the results has several important flaws as described in the major comments. Authors also suggested higher adaptability of legume tree species to low light environment. However, the suggestion does not sound convincing because of the flaws in the interpretation of the results and the small number of species used in the study.
Major comments
- Authors suggested that “the adaptability of N-fixing species to low light environment is better than that of non-N-fixing species” because “the photosynthetic N use efficiency (PNUE) of the two N-fixing trees was higher under 10% irradiance treatment” in Abstract and Conclusion. However, PNUE is calculated as Asat/Narea, whrere Asat is a photosynthesis rate at saturating light condition and Narea is nitrogen content per leaf area (line 300), and is not a parameter showing the ability of carboxylation in the low light condition. In low light condition, the light intensity rarely goes to the saturating light intensity, therefore, high Asat is not necessary for plants. If authors would like to discuss about the “adaptability of N-fixing species to low light environment”, authors need to discuss with the parameters which indicates the photosynthetic ability in the low light condition, such as daily carbon gain per nitrogen content or apparent quantum yield (initial slope of the light-photosynthesis curve).
- Authors mentioned that “decreased Asat was main reason for the decrease in their gm” (line 191). But this logic is wrong. Asat: assimilation is a result of physiological photosynthetic-capacity (such as Vcmax, Jmax) and physical conductance of CO2 (such as gs and gm). Previous studies showed that leaf anatomical difference, which can be changed among light condition, significantly affects the mesophyll conductance: gm (von Caemmerer & Evans 1991 Australian Journal of Plant Physiology 18, 287–305.; Oguchi et al. 2003 Plant Cell and Environment 26, 505-512; Terashima et al. 2006 Journal of Experimental Botany 57, 343-354 etc.). If authors would like to discuss about the mesophyll conductance, authors should consider about the anatomical difference of leaf traits among species in the 3 different light conditions, but unfortunately the data of leaf anatomical traits are lacked.
- Although authors discuss about the ability of light harvesting in lines 211-213, there is almost no analyses about the difference of photosynthetic rate in low light condition (e.g. apparent quantum yield). Therefore, the discussion does not sound convincing.
- The methodological information of Figure 3 is almost totally lacked. Did authors used a type of path analysis? What does “Transfer to” in the figure mean? How the “results in” relationships were analyzed? The methods should be clearly described.
- If the data of high light intensity in the present manuscript is the same data from the reference 28 (Tang et al. 2019 Plos one) (and high nitrogen condition of reference 29 (Tang et al. 2019 Scientific report)), it should be clearly described. The discussion about the knowledge gained in these three related papers would be also needed.
- Lines 166-168: Why is it that non N-fixing tress would not have adequate N concentrations in leaves?
- Although I suggested some English revisions in my minor comments below, there are more points needed to be revised. I think English editing is necessary throughout the text.
Minor comments
- Line 43: "carbon assimilation" or "carbon fixation" would be better than "carbon reaction".
- Lines 54-56: English of this sentence is awkward (such as “significant effect” should be “significantly affect"), and need a revision.
- Lines 217-218: This sentence does not make sense. Need a revision.
- Line 234: How do authors decide “unimportant N”? Such N may have some functions. “Unimportant” can be also changed depending on readers, which makes this sentence blur. Authors need to explain it with its functions.
- Lines 269-270: Does this sentence mean that “Respiration rate” was measured?
- Line 271: light intensity of saturated PPFD should be concretely described. “An were measured too” seems to be overlapped with the previous sentences.
- Line 315: Explanations of Vcr, Jmc and CB are necessary. A brief explanation of the method of calculation should be also necessary.
Author Response
Dear reviewer
Re: Manuscript ID: plants-1377753
Please find attached a revised version of our manuscript “The effect of low irradiance on leaf nitrogen allocation and mesophyll conductance to CO2 in seedlings of four tree species in subtropical China”, which we would like to resubmit for publication as a research paper in Plants.
Your comments were highly insightful and enabled us to greatly improve the quality of our manuscript. In the following pages are our point-by-point responses to each of the comments of the reviewers as well as your own comments.
Revisions in the text are shown using yellow highlight for additions, and strikethrough font [example] for deletions. We hope that the revisions in the manuscript and our accompanying responses will be sufficient to make our manuscript suitable for publication in Plants.
We shall look forward to hearing from you at your earliest convenience.
Thank you and best regards.
Yours sincerely,
Jingchao Tang
Ph: (+86)18866482400
E–mail: tjc@qut.edu.cn
Responses to the comments of Reviewer #2
- Major comments
Question 1:. Authors suggested that “the adaptability of N-fixing species to low light environment is better than that of non-N-fixing species” because “the photosynthetic N use efficiency (PNUE) of the two N-fixing trees was higher under 10% irradiance treatment” in Abstract and Conclusion. However, PNUE is calculated as Asat/Narea, whrere Asat is a photosynthesis rate at saturating light condition and Narea is nitrogen content per leaf area (line 300), and is not a parameter showing the ability of carboxylation in the low light condition. In low light condition, the light intensity rarely goes to the saturating light intensity, therefore, high Asat is not necessary for plants. If authors would like to discuss about the “adaptability of N-fixing species to low light environment”, authors need to discuss with the parameters which indicates the photosynthetic ability in the low light condition, such as daily carbon gain per nitrogen content or apparent quantum yield (initial slope of the light-photosynthesis curve).
Response: You are right, and we added four leaf traits- dark respiration (Rn, μmol m–2 s–1), light saturation point (LSP μmol m–2 s–1), light compensation point (LCP μmol·m–2·s–1) and apparent quantum yield (AQY mol mol–1) to discuss about the adaptability of N-fixing species to low light environment. We added a table (new Table 5) to show data of these traits, and described and discussed these traits
[Line 341 to 349 in revised version]
The photosynthetic response to the photosynthetic photon flux density (PPFD, µmol m-2 s-1) was determined under a leaf chamber CO2 concentration of 380 μmol mol–1, and the net photosynthetic rate (An, μmol m–2 s–1), CO2 concentration at sub-stomatal cavities (Ci, μmol mol–1), and stomatal conductance (gs, mol CO2 m–2 s–1) were measured at photon flux densities of 1500, 1200, 1000, 800, 600, 400, 200, 150, 100, 80, 50, 30, 20, 10, and 0 μmol m–2 s–1 (see Fig. S1). The PPFD-saturated net CO2 assimilation rate (Asat, μmol m–2 s–1), dark respiration (Rn, μmol m–2 s–1), LSP (μmol m–2 s–1), and LCP (μmol m–2 s–1) were then measured from the light response curves. The AQY (mol mol–1) was measured as the initial slope of the light response curves (PPFD ≤ 200 μmol m–2 s–1).
[Line 166 to 182 in revised version]
The apparent quantum yield (AQY) and Rn of D. odorifera, C. hystrix, and B. alnoides were not significantly affected by low-irradiance treatment; however, the AQY of E. fordii under the 10% and 40% irradiance treatments were significantly higher than that under 100% irradiance, and Rn of E. fordii under the 10% and 40% irradiance treatments were significantly lower than those under 100% irradiance (Table 5). The light compensation point (LCP) of D. odorifera and C. hystrix under the 10% and 40% irradiance treatments, and E. fordii and B. alnoides under the 10% irradiance treatment were significantly lower than those under 100% irradiance (Table 5). The light saturation point (LSP) of D. odorifera and E. fordii under the 10% irradiance treatment, and C. hystrix and B. alnoides under the 10% and 40% irradiance treatments were significantly lower than those under 100% irradiance (Table 5).
Table 5. Apparent quantum yield (AQY), dark respiration (Rn), light compensation point (LCP) and light saturation point (LSP) in Dalbergia odorifera, Erythrophleum fordii, Betula alnoides, and Castanopsis hystrix grown under three different irradiance treatments. Data are means of seven plants per treatment ± SE. Lower case letters indicate significant difference at 0.05 levels among the irradiance treatments, whereas capital letters indicate significant difference at 0.05 levels among species under same irradiance treatment. F-ratios with statistically significant values denoted by *P < 0.05, **P < 0.01, ***P < 0.001 among irradiance treatment.
Tree species |
Irradiance treatment |
AQY (mol mol–1) |
Rn (μmol m–2 s–1) |
LCP(μmol m–2 s–1) |
LSP(μmol m–2 s–1) |
Dalbergia Odorifera |
100% |
0.049 ± 0.004aA |
0.909 ± 0.050aBC |
22.1 ± 1.68aA |
822.9 ± 27.5aA |
40% |
0.055 ± 0.001aA |
0.845 ± 0.050aA |
14.4 ± 0.73bB |
724.3 ± 37.0abA |
|
10% |
0.053 ± 0.002aAB |
0.760 ± 0.038aB |
7.4 ± 0.43cB |
684.3 ± 23.1bA |
|
F |
1.146 |
2.587 |
46.127*** |
5.378* |
|
Erythrophleum fordii |
100% |
0.045 ± 0.002bA |
1.129 ± 0.051aA |
13.9 ± 0.81aB |
637.1 ± 29.6aB |
40% |
0.057 ± 0.001aA |
0.873 ± 0.050bA |
13.1 ± 1.10aB |
633.6 ± 17.1aA |
|
10% |
0.055 ± 0.002aA |
0.936 ± 0.030bAB |
7.5 ± 0.79bB |
522.1 ± 17.2bB |
|
F |
11.735** |
8.811** |
14.464*** |
6.569** |
|
Castanopsis hystrix |
100% |
0.045 ± 0.003aA |
1.005 ± 0.067aAB |
14.0 ± 1.21aB |
632.9 ± 23.4aB |
40% |
0.048 ± 0.001aA |
0.988 ± 0.040aA |
7.3 ± 1.00bC |
307.1 ± 26.8bC |
|
10% |
0.047 ± 0.003aAB |
1.048 ± 0.088aA |
7.4 ± 1.21bB |
262.1 ± 27.7bC |
|
F |
0.352 |
0.206 |
11.226** |
60.359*** |
|
Betula alnoides |
100% |
0.047 ± 0.002aA |
0.889 ± 0.039aBC |
20.8 ± 0.93aA |
886.4 ± 43.5aA |
40% |
0.055 ± 0.006aA |
0.844 ± 0.055aA |
18.6 ± 2.49aA |
519.3 ± 27.9bB |
|
10% |
0.044 ± 0.003aB |
0.834 ± 0.048aAB |
10.5 ± 0.94bA |
268.6 ± 30.5cC |
|
F |
1.988 |
0.384 |
10.989** |
80.343*** |
[Line 261 to 276 in revised version]
As these species are commonly used to plant in gaps or mixed with other species, their tolerance to low light conditions will affect the growth effect after planting. All four species decreased the LMA to increase the area receiving light (Table 1) [1,4,8,14], increased PL to increase their light-harvesting capacity and sustain photosynthesis (Table 4) [3–4], and decreased the LCP and LSP to increase the ability to use low light (Table 5) under low light conditions. Meanwhile, N-fixing plants exhibited some other adaptations to low light conditions, such as increased Ci and Cc, maintained Asat under the 10% and 40% irradiance treatments (Tables 1 and 2), and increased PNUE under the 10% and 40% irradiance treatments (Table 1). N-fixing plants also had higher Asat, Narea, Vcmax, and Jmax than non-N-fixing species under the 10% irradiance treatment (Table 1,3). AQY refers to the ability to use low light [43] E. Fordii exhibited improved AQY under the 10% and 40% irradiance treatments, and the AQY values under the 10% irradiance treatment in the order of: E. fordii > D. odorifera > C. hystrix > B. Alnoides (Table 5). E. fordii also reduced Rn under the 10% and 40% irradiance treatments to reduce respiratory expenditure (Table 5). In conclusion, these results suggest that the adaptability of N-fixing species to low light environments is better than that of non-N-fixing species.
Question 2:. Authors mentioned that “decreased Asat was main reason for the decrease in their gm” (line 191). But this logic is wrong. Asat: assimilation is a result of physiological photosynthetic-capacity (such as Vcmax, Jmax) and physical conductance of CO2 (such as gs and gm). Previous studies showed that leaf anatomical difference, which can be changed among light condition, significantly affects the mesophyll conductance: gm (von Caemmerer & Evans 1991 Australian Journal of Plant Physiology 18, 287–305.; Oguchi et al. 2003 Plant Cell and Environment 26, 505-512; Terashima et al. 2006 Journal of Experimental Botany 57, 343-354 etc.). If authors would like to discuss about the mesophyll conductance, authors should consider about the anatomical difference of leaf traits among species in the 3 different light conditions, but unfortunately the data of leaf anatomical traits are lacked.
Response: You are right, the logic in this place was wrong, and we have rewritten this passage. The data of leaf anatomical traits are lacked indeed, and we use LMA to discussed about the relationship between anatomical difference and mesophyll conductance. Of course, the discussion here is a little bit simple, but we will obtain the data of leaf anatomical traits in further studies.
[Line 235 to 244 in revised version]
gm in D. odorifera, C. hystrix, and B. alnoides seedlings decreased under 10% irradiance, which was consistent with previous studies [19–20] (Table 2). gm could be affected by leaf anatomical differences, such as cell wall thickness, surface area of mesophyll cells, number of mesophyll layers, and leaf stomata density [17,18]. Variations in LMA could be driven by several anatomical traits, such as the cell wall thickness and number of mesophyll layers [34], and changes in LMA always influence gm [35]. If a lower LMA is the result of mesophyll cell wall thinning, it will increase gm [36–37]; if it is the result of a lower number of mesophyll layers, it will decrease gm [38]. In this study, the LMA of D. odorifera, C. hystrix, and B. alnoides decreased under the 10% irradiance treatment, indicating that low light may decrease the number of mesophyll layers in these tree seedlings.
Question 3:. Although authors discuss about the ability of light harvesting in lines 211-213, there is almost no analyses about the difference of photosynthetic rate in low light condition (e.g. apparent quantum yield). Therefore, the discussion does not sound convincing.
Response: We added four leaf traits- dark respiration (Rn, μmol m–2 s–1), light saturation point (LSP μmol m–2 s–1), light compensation point (LCP μmol·m–2·s–1) and apparent quantum yield (AQY mol mol–1) to discuss about the adaptability of N-fixing species to low light environment. We added a table (new Table 5) to show data of these traits, and described and discussed these traits. Please see our Response to Question 1 for more details.
Question 4:. The methodological information of Figure 3 is almost totally lacked. Did authors used a type of path analysis? What does “Transfer to” in the figure mean? How the “results in” relationships were analyzed? The methods should be clearly described.
Response: We added methodological information of Figure 3 in our manuscript.
[Line 305 to 311 in revised version]
Fig. 3. Leaf trait changes in Dalbergia odorifera, Erythrophleum fordii, Betula alnoides, and Castanopsis hystrix grown under 10% irradiance versus 100% irradiance treatments. “Red upward arrow” means this leaf trait under 10% irradiance treatment was significant higher than 100%, “Blue down arrow” means this leaf trait under 10% irradiance treatment was significant lower than 100%, and “green rectangle” means this leaf trait under 10% irradiance treatment was no significant change with 100%. Specific data information and significance see Tables 1-4. The leaf trait pointed at by “Thin blue arrow” can be affected by the traits at the other end of arrow, and the increased part of leaf trait pointed at by “Thin red arrow” might come from the reduced part of leaf traits at the other end of arrow.
Question 5:. If the data of high light intensity in the present manuscript is the same data from the reference 28 (Tang et al. 2019 Plos one) (and high nitrogen condition of reference 29 (Tang et al. 2019 Scientific report)), it should be clearly described. The discussion about the knowledge gained in these three related papers would be also needed.
Response: We described the data of high light in the present manuscript is same data from the reference 28 (Tang et al. 2019 Plos one), and high nitrogen condition of reference 29 (Tang et al. 2019 Scientific report). We also discussed about the knowledge gained in these three related papers.
[Line 277 to 286 in revised version]
We previously studied the interspecific differences between D. odorifera and E. fordii (N-fixing trees), and C. hystrix and B. alnoides (non-N-fixing trees) [44], and how they are affected by soil N deficiencies [45]. The data obtained under high light intensity in this manuscript are the same as those used by Tang et al. [44] and the high nitrogen condition reported by Tang et al. [45], which were used as the “Control group.” In [44], N-fixing trees had higher Narea and Nmass, but lower PR, PB, and PNUE than non-N-fixing trees. In [45], soil N deficiency had less influence on the leaf N concentration and photosynthetic ability in the two N-fixing trees. Combined with the results of this study, we consider that nitrogen-fixing plants are suitable species for afforestation, and could be independently planted in poor soil, mixed with non-N-fixing species, or planted in gaps.
Question 6:. Lines 166-168: Why is it that non N-fixing tress would not have adequate N concentrations in leaves?
Response: What I'm trying to say in this sentence was “We hypothesized that N-fixing trees could fix nitrogen from the air, Therefore, the reduction of Narea under low light treatment may be smaller than that of non-N-fixing tree species”, we have modified the original sentence.
[Line 210 to 218 in revised version]
As Narea=LMA×Nmass, the significant decrease in LMA led to a decrease in Narea under the 10% and 40% irradiance treatments, indicating that thinner leaves had a lower concentration of N per unit area [1,10,30]. We hypothesized that N-fixing trees would have adequate N concentrations in leaves, and that the lower proportion of Narea under low light treatment may be smaller in N-fixing than in non-N-fixing trees. We hypothesized that N-fixing trees could fix nitrogen from the air; therefore, the reduction in Narea under low light may be smaller than that of non-N-fixing tree species. However, our results indicate that the decrease in the proportion of Narea was not lower in N-fixing trees than that in non-N-fixing trees (D. odorifera: -55.70%, E. fordii: -22.39%, C. hystrix:-22.54%, and B. alnoides: -45.63%). The N fixation capacities of D. odorifera and E. fordii did not limit the reduction in Narea under low light treatment.
Question 7:. Although I suggested some English revisions in my minor comments below, there are more points needed to be revised. I think English editing is necessary throughout the text.
Response: Our manuscript was edited by professional editors at Editage to improve English, please see attachment for proof materials
Minor comments
Question 1:. Line 43: "carbon assimilation" or "carbon fixation" would be better than "carbon reaction".
Response: You are right, and we changed "carbon reaction" to "carbon assimilation"
[Line 43 to 46 in revised version]
and some plants may also change the fraction of leaf N allocated to Rubisco (PR) and bioenergetics (PB) to balance the light reaction with carbon assimilation and achieve optimal photosynthetic efficiency [8–9].
Question 2: Lines 54-56: English of this sentence is awkward (such as “significant effect” should be “significantly affect"), and need a revision.
Response: You are right, and have rewritten this sentence.
[Line 56 to 58 in revised version]
Low irradiance decreased gm [19–20] or did not significantly affect gm in different species [21–22]. Therefore, changes in gm in different species should be further studied.
Question 3:. Lines 217-218: This sentence does not make sense. Need a revision.
Response: You are right, and we deleted this sentence.
[Line 258 to 260 in revised version]
However, our results indicated that the effect of gm on PNUE was not significant under varying light treatments in all tree species (Fig 2). Under changing irradiance treatments, these four tree species did not adjust their allocation of N, which could influence gm and affect PNUE.
Question 4:. Line 234: How do authors decide “unimportant N”? Such N may have some functions. “Unimportant” can be also changed depending on readers, which makes this sentence blur. Authors need to explain it with its functions.
Response: You're right, “unimportant N” is ambiguous here, so we've rewritten this sentence.
[Line 300 to 302 in revised version]
These different strategies are related to the ecological characteristics of each tree species, but the goal is the same (reducing unimportant some other N components and increase important N components (e.g. PL) light-harvesting N components under low light treatment).
Question 5:. Lines 269-270: Does this sentence mean that “Respiration rate” was measured?
Response: You are right, we measured respiratory rate from light response curves indeed, and we have rewritten these sentences to make our presentation much clearer.
[Line 341 to 349 in revised version]
The photosynthetic response to the photosynthetic photon flux density (PPFD, µmol m-2 s-1) was determined under a leaf chamber CO2 concentration of 380 μmol mol–1, and the net photosynthetic rate (An, μmol m–2 s–1), CO2 concentration at sub-stomatal cavities (Ci, μmol mol–1), and stomatal conductance (gs, mol CO2 m–2 s–1) were measured at photon flux densities of 1500, 1200, 1000, 800, 600, 400, 200, 150, 100, 80, 50, 30, 20, 10, and 0 μmol m–2 s–1 (see Fig. S1). The PPFD-saturated net CO2 assimilation rate (Asat, μmol m–2 s–1), dark respiration (Rn, μmol m–2 s–1), LSP (μmol m–2 s–1), and LCP (μmol m–2 s–1) were then measured from the light response curves. The AQY (mol mol–1) was measured as the initial slope of the light response curves (PPFD ≤ 200 μmol m–2 s–1).
Question 6:. Line 271: light intensity of saturated PPFD should be concretely described. “An were measured too” seems to be overlapped with the previous sentences.
Response: We added CO2 concentration and light gradients about the CO2 and light response curve in our manuscript.
[Line 337 to 356 in revised version]
The experiment was conducted between 09:00 and 11:00 h on sunny days, on newly fully expanded leaves of seven seedlings per treatment from July to August 2014, lasting for two months. An LI-6400-40 portable photosynthesis system (LI-COR, Lincoln, Nebraska, USA) was used to determine the photosynthetic light and CO2 response curves. The photosynthetic response to the photosynthetic photon flux density (PPFD, µmol m-2 s-1) was determined under a leaf chamber CO2 concentration of 380 μmol mol–1, and the net photosynthetic rate (An, μmol m–2 s–1), CO2 concentration at sub-stomatal cavities (Ci, μmol mol–1), and stomatal conductance (gs, mol CO2 m–2 s–1) were measured at photon flux densities of 1500, 1200, 1000, 800, 600, 400, 200, 150, 100, 80, 50, 30, 20, 10, and 0 μmol m–2 s–1 (see Fig. S1). The PPFD-saturated net CO2 assimilation rate (Asat, μmol m–2 s–1), dark respiration (Rn, μmol m–2 s–1), LSP (μmol m–2 s–1), and LCP (μmol m–2 s–1) were then measured from the light response curves. The AQY (mol mol–1) was measured as the initial slope of the light response curves (PPFD ≤ 200 μmol m–2 s–1).
The CO2 response curve was determined under saturated PPFD, and Ci and gs were measured under leaf chamber CO2 concentrations of 380, 200, 150, 100, 80, 50, 380, 600, 800, 1000, 1200, 1500, 1800, and 2000 μmol mol–1 (see Fig. S2). The light- and CO2-saturated net CO2 assimilation rate (Amax, μmol m–2 s–1) was then measured from the CO2 response curve. The fluorescence yield (⊿F/Fm) was measured under leaf chamber CO2 concentrations of 380 μmol mol–1 and saturated PPFD. Meanwhile, the relative humidity of the leaf chamber was maintained at 50 ± 5%, and the leaf temperature was maintained at 25 ± 2 °C.
Question 7:
Line 315: Explanations of Vcr, Jmc and CB are necessary. A brief explanation of the method of calculation should be also necessary.
Response: You're right, and we added explanations and calculation method of Vcr, Jmc and CB in this part.
[Line 393 to 405 in revised version]
where CChl is the chlorophyll concentration (mmol g–1), Vcr is the specific activity of Rubisco (μmol CO2 g–1 Rubisco s–1), Jmc is the potential rate of photosynthetic electron transport (μmol electrons μmol–1 Cyt f s–1), and CB is the ratio of leaf chlorophyll to leaf nitrogen during light-harvesting (mmol Chl (g N)–1). Vcr, Jmc, and CB were calculated according to Niinemets and Tenhunen [56]:
................................................................. (6)
....................................................................(7)
where R is the gas constant (8·314 J K–1 mol–1), Tk is the leaf temperature (K), ΔHa is the activation energy, ΔHd is the deactivation energy, ΔS is the entropy term, and c is the scaling constant. [LMA] and [CB] are the values of LMA and CB, respectively. The values of ΔHa, ΔHd, ΔS, and c were 74000 J·mol-1, 203000 J·mol-1, 645 J·K–1·mol–1, and 32.9 when calculating Vcr, and 24100 J·mol-1, 564150 J·mol-1, 1810 J·K–1·mol–1, and14.77 when calculating Jmc [56].

Reviewer 3 Report
The manuscript looks to know if the nitrogen allocation in the leaf photosynthetic machinery at low irradiancies is different in nitrogen fixing plants and in non-fixing plants. As expected, adaptation to low light increases the percentage of nitrogen compromised in light-harvesting complexes and decreases it in Rubisco. The adaptation to low light decreases photosynthetic activity in Castanopsis hystrix and Betula alnoides (non-N-fixing trees) more than in Dalbergia odorifera and Erythrophleum fordii (N-fixing trees). OK.
The assays and measurements are well designed, and the results have interest although they mainly confirm current paradigms of the relation between carbon and nitrogen metabolism in plants and, more specifically, in the leaves. However, frequent grammatical mistakes and deficient description of some experiment details must be corrected.
Sometimes, poor English results in incorrect scientific statements that must be corrected. E. g.:
In line 36, “to synthesize water and CO2 into carbohydrates” is not correct and could be changed to “to synthesize carbohydrates from water and CO2” or something similar.
In following line 37 (that repeat the same of line 17) “lack of light can lead to a decrease”. Without light there is no photosynthetic activity. One suggestion: change “lack of light” to “low light intensity”.
Lines 67-69, delete …”, and”… in line 68.
Misleading grammars, as those are frequent and must be carefully corrected. The English must be corrected by expert.
A few experimental points must be detailed:
- For the Tables. Define what F-ratio is. Preferent at Materials and Methods.
- When were photosynthesis measurements carried out? By end June? How much irradiation treatment last? What is the time course of leaf adaptation to low light?
Author Response
Dear reviewer
Re: Manuscript ID: plants-1377753
Please find attached a revised version of our manuscript “The effect of low irradiance on leaf nitrogen allocation and mesophyll conductance to CO2 in seedlings of four tree species in subtropical China”, which we would like to resubmit for publication as a research paper in Plants.
Your comments were highly insightful and enabled us to greatly improve the quality of our manuscript. In the following pages are our point-by-point responses to each of the comments of the reviewers as well as your own comments.
Revisions in the text are shown using yellow highlight for additions, and strikethrough font [example] for deletions. We hope that the revisions in the manuscript and our accompanying responses will be sufficient to make our manuscript suitable for publication in Plants.
We shall look forward to hearing from you at your earliest convenience.
Thank you and best regards.
Yours sincerely,
Jingchao Tang
Ph: (+86)18866482400
E–mail: tjc@qut.edu.cn
Responses to the comments of Reviewer #3
Question 1:. The assays and measurements are well designed, and the results have interest although they mainly confirm current paradigms of the relation between carbon and nitrogen metabolism in plants and, more specifically, in the leaves. However, frequent grammatical mistakes and deficient description of some experiment details must be corrected. Sometimes, poor English results in incorrect scientific statements that must be corrected. E. g.:
Response: Our manuscript was edited by professional editors at Editage to improve English, please see attachment for proof materials
Question 2:. In line 36, “to synthesize water and CO2 into carbohydrates” is not correct and could be changed to “to synthesize carbohydrates from water and CO2” or something similar.
Response: You are right, and we modified this sentence
[Line 36 to 38 in revised version]
Radiation is a source of energy for plants. Through photosynthesis, green plants use light to synthesize carbohydrates from water and CO2, which are necessary for maintaining growth and development.
Question 3:. In following line 37 (that repeat the same of line 17) “lack of light can lead to a decrease”. Without light there is no photosynthetic activity. One suggestion: change “lack of light” to “low light intensity”.
Response: You are right, and we modified these parts
[Line 17 in revised version]
Abstract: A lack of light Low light intensity can lead to a decrease in photosynthetic capacity.
[Line 39 to 41 in revised version]
A lack of light Low light intensity can lead to a decrease in photosynthetic capacity, forcing plants to change their leaf photosynthesis system and structure to increase their light-harvesting ability [2–5].
Question 4:. Lines 67-69, delete …”, and”… in line 68.
Response: You are right, and we rewrite this sentence.
[Line 66 to 71 in revised version]
However, few studies have been conducted on whether low-irradiance treatment can affect the PNUE of N-fixing trees and the relevant internal control mechanism of leaf N allocation and gm. We suspect that N-fixing species with sufficient N in their leaves could increase their PR, PB, and PL to increase the PNUE under low-irradiance treatment, and maintain their photosynthetic capacity and growth better than non-N-fixing species under low-irradiance environments.
Question 5:. Misleading grammars, as those are frequent and must be carefully corrected. The English must be corrected by expert.
Response: Our manuscript was edited by professional editors at Editage to improve English, please see attachment for proof materials
Question 6:. A few experimental points must be detailed: For the Tables. Define what F-ratio is. Preferent at Materials and Methods.
Response: We defined what F-ratio is at Materials and Methods.
[Line 412 to 418 in revised version]
The differences between the four seedling species and different irradiance treatments were analyzed using one-way analysis of variance (ANOVA), and a post-hoc test (Tukey’s test) was conducted to determine if the differences were significant. The F-ratio in the tables is the ratio of the mean squares between groups and within groups, and P is the confidence interval of F. The significance of the linear relationships between each pair of variables was tested by Pearson’s correlation (two-tailed). All analyses were conducted using Statistical Product and Service Solutions 17.0 (version 17.0; SPSS, Chicago, IL, USA).
Question 7:. When were photosynthesis measurements carried out? By end June? How much irradiation treatment last? What is the time course of leaf adaptation to low light?
Response: Determination of gas exchange and fluorescence parameters was carried out from July to August, lasted for two months. The irradiation treatment (100% 40% and 10% of sunlight irradiance) was carried out from April to June, lasted for three months. We added these details in “Materials and Methods”.
[Line 323 to 326 in revised version]
From April to June 2014, three levels of irradiance, that is, 100%, 40%, and 10% of sunlight irradiance, were applied using neutral black polypropylene frames with a covering film of black polyolefin resin fine mesh. The irradiation treatment lasted for three months.
[Line 337 to 339 in revised version]
The experiment was conducted between 09:00 and 11:00 h on sunny days, on newly fully expanded leaves of seven seedlings per treatment from July to August 2014, lasting for two months.

Reviewer 4 Report
See the file attached.

Author Response
Dear reviewer
Re: Manuscript ID: plants-1377753
Please find attached a revised version of our manuscript “The effect of low irradiance on leaf nitrogen allocation and mesophyll conductance to CO2 in seedlings of four tree species in subtropical China”, which we would like to resubmit for publication as a research paper in Plants.
Your comments were highly insightful and enabled us to greatly improve the quality of our manuscript. In the following pages are our point-by-point responses to each of the comments of the reviewers as well as your own comments.
Revisions in the text are shown using yellow highlight for additions, and strikethrough font [example] for deletions. We hope that the revisions in the manuscript and our accompanying responses will be sufficient to make our manuscript suitable for publication in Plants.
We shall look forward to hearing from you at your earliest convenience.
Thank you and best regards.
Yours sincerely,
Jingchao Tang
Ph: (+86)18866482400
E–mail: tjc@qut.edu.cn
Responses to the comments of Reviewer #4
Question 1:. In Table 1, the data on PNUE are given with 4 significant digits with accuracy of 0.01 μmol m-2s-1. Are the data so accurate? How about other data such as Nmass and LMA? In Table 2, the data on Ci. Cc and (Ci.-Cc) are given with 4 or 5 significant digits with accuracy of 0.01 umol m-2s-1. What is the accuracy of CO2 concentration analyzer used in the experiment? The same question regarding Jmax in Table 3
Response: You are right, the accuracy of data on leaf traits you mentioned were not suitable, and we modified the accuracy of data on these leaf traits.
[Line 108 in revised version]
Nmass (mg g-1) |
LMA (g m-2) |
PNUE (μmol mol–1 s–1) |
31.7 ± 0.76aA |
69.0 ± 3.90aB |
52.6 ± 3.78bB |
31.2± 0.65aA |
51.8 ± 0.65bB |
72.3 ± 7.03bB |
33.0 ± 1.11aA |
29.3 ± 0.67cC |
101.0 ± 7.12aA |
1.196 |
73.752*** |
15.533*** |
28.1 ± 1.49bB |
71.4 ± 0.89aB |
45.9 ± 2.24cB |
33.0 ± 0.46bA |
53.1 ± 0.99bB |
75.0 ± 4.56aB |
35.3 ± 0.88aA |
44.3 ± 1.47cB |
61.6 ± 3.72bB |
12.658*** |
145.227*** |
15.877*** |
10.2 ± 1.80bD |
100.1 ± 2.60aA |
112.0 ± 4.62aA |
9.6 ± 0.50bC |
78.8 ± 1.11bA |
87.0± 7.26bB |
13.7 ± 0.49aC |
57.9 ± 1.29cA |
74.4 ± 4.59bB |
28.220*** |
138.877*** |
12.868*** |
15.4 ± 1.04bC |
67.6 ± 5.45aB |
120.5 ± 5.18abA |
15.4 ± 0.45bB |
49.1 ± 3.36bB |
140.3 ± 8.02aA |
19.0 ± 0.62aB |
29.6 ± 2.14cC |
105.3 ± 8.33bA |
7.790** |
23.833*** |
3.815* |
[Line 130 in revised version]
Ci (μmol mol-1) |
Cc (μmol mol-1) |
Ci-Cc (μmol mol-1) |
251.5 ± 6.44bBC |
190.8±6.92bB |
60.8±2.21aC |
288.5±3.93aA |
210.0±8.82bA |
78.6±7.50aAB |
302.6±1.94aA |
231.3±6.20aA |
71.2±5.27aB |
34.333*** |
7.536** |
2.698 |
235.6 ± 6.19bC |
132.6±6.90bD |
103.0±4.83aA |
254.1±3.81abB |
156.8±4.09aB |
97.3±2.37aA |
264.4±4.18aB |
171.7±4.45aB |
92.7±1.56aA |
7.982** |
13.855*** |
2.553 |
256.8 ± 5.24bB |
168.0±6.04bC |
88.8±6.26aB |
280.1±3.48aA |
196.4±5.84aA |
83.7±4.14aAB |
268.0±5.65abB |
186.5±6.27abB |
81.5±5.03aAB |
5.711* |
5.691* |
0.511 |
292.9 ± 5.94aA |
226.4±9.57aA |
66.5±4.64aC |
297.1±7.04aA |
222.9±12.19aA |
74.2±5.69aB |
312.3±4.73aA |
232.1±9.07aA |
80.3±6.14aAB |
2.93 |
0.198 |
1.568 |
[Line 138 in revised version]
Vcmax (μmol m–2 s–1) |
Jmax (μmol m–2 s–1) |
78.1±4.59bB |
100.7±5.80aBC |
95.2±8.01aB |
118.5±7.39aB |
68.6±3.96cB |
79.1±2.76bB |
5.405* |
12.154*** |
99.8±9.37bA |
128.8±11.20bAB |
141.4±5.24aA |
168.9±3.36aA |
80.1±4.07cA |
99.8±3.83cA |
22.233*** |
23.930*** |
82.8±4.47aB |
109.3±3.40aABC |
46.5±2.51bC |
57.6±4.49bC |
47.7±2.92bC |
66.5±3.80bC |
75.031*** |
49.677*** |
73.0±3.51aB |
98.4±5.37aBC |
82.6±5.46aB |
97.8±5.39aB |
41.6±4.80bC |
56.0±4.59bC |
25.05*** |
22.465*** |
Question 2:. The term 'irradiance' (or radiation energy flux density) should be is used with unit of W m-2 (or J m-2s-1). The term PPFD (photosynthetic photon flux density) is used with unit of μmol m-2s-1. In case that the PPFD meter was used in the experiment and the data were expressed in unit of μmol m-2s-1, the terms irradiance and irradiance treatment should not be used. Instead, the terms PPFD and PPFD treatment should be used. PPFD value is not proportional to irradiance value, because the energy per photon increases with increasing the wavelength. The title of the paper should be The effect of low PPFD on
Response: We found a mistake when checking description of this part, MT-4617LED-C monochromator spectroradiometer (Pro's Kit Ltd, Shanghai, China) is a kind of spectroradiometer, only could get light irradiance with unit of “Lux”. We convert unit of “Lux” to “μmol·m-2·s-1” by “1K Lux=18 μmol·m-2·s-1” when the light source is daytime sunlight, and this can lead to misunderstanding about our experiment. So, we have revised the original description in “Materials and Methods”.
[Line 326 to 329 in revised version]
Photosynthetic photon flux density (PPFD) Illumination was measured using an MT-4617LED-C monochromator spectroradiometer (Pro's Kit Ltd, Shanghai, China); the average sunny midday PPFD illumination in the 100%, 40%, and 10% irradiance treatments were 1400, 560 and 140μmol m−2 s−1,78000, 31000, and 7800 lux, respectively.
Question 3:. In the experiment, three levels of PPFD were applied using neutral black polypropylene frames. Under such treatments, air temperature, relative humidity and air movement under frames are also often affected by the degree of shading. Describe the type of covering film.
Response: The type of covering film was black polyolefin resin fine mesh, and we added these details in “Materials and Methods”.
[Line 323 to 325 in revised version]
three levels of irradiance, that is, 100%, 40%, and 10% of sunlight irradiance, were applied using neutral black polypropylene frames with a covering film of black polyolefin resin fine mesh.

Round 2
Reviewer 1 Report
The authors replied thoroughly to all the Reviewers' comments and now the paper quality is significantly improved.
Author Response
Dear reviewer
Re: Manuscript ID: plants-1377753
Please find attached a revised version of our manuscript “The effect of low irradiance on leaf nitrogen allocation and mesophyll conductance to CO2 in seedlings of four tree species in subtropical China”, which we would like to resubmit for publication as a research paper in Plants.
Your comments were highly insightful and enabled us to greatly improve the quality of our manuscript. In the following pages are our point-by-point responses to each of the comments of the reviewers as well as your own comments.
Revisions in the text are shown using yellow highlight for additions, and strikethrough font [example] for deletions. We hope that the revisions in the manuscript and our accompanying responses will be sufficient to make our manuscript suitable for publication in Plants.
We shall look forward to hearing from you at your earliest convenience.
Thank you and best regards.
Yours sincerely,
Jingchao Tang
Ph: (+86)18866482400
E–mail: tjc@qut.edu.cn
Responses to the comments of Reviewer #1
Question 1: The authors replied thoroughly to all the Reviewers' comments and now the paper quality is significantly improved.
Response1: Thank you very much for your guidance on this article, your comments were highly insightful and enabled us to greatly improve the quality of our manuscript.

Reviewer 2 Report
Major comments
- Authors used the photosynthetic data under PPFD ≤ 200 umol m-2 s-1 for the calculation of apparent quantum yield (AQY). However, the light intensity would be too strong that the photosynthetic rate could not be fitted by linear fitting (as we can see in Fig. S1). Because authors measured photosynthetic rate at 0, 10, 20, 30, 50 umol m-2 s-1 and more light intensities, the photosynthetic data under PPFD ≤ 30 or 50 umol m-2 s-1 would be enough for the statistically robust calculation of AQY. The results will be largely changed.
- As authors mentioned in line 186, this study discusses about the PNUE under low-irradiance treatment in many places. However, as I mentioned in the previous comments, PNUE in this study only uses Asat (photosynthesis rate at saturating light intensity) for the calculation, which does not indicate the adaptability of the plants to low light condition. (Authors currently just using the data of PNUE at saturating light intensity of plants grown under low light: 10% light condition.) Authors should use, for example, photosynthesis rate at 100 umol m-2 s-1 (which is in the range of the growth irradiance in the 10% light condition), for the calculation of PNUE (which can be called as PNUE100 and so on). Otherwise, the almost all analyses about PNUE in the current manuscript does not indicate the adaptability of the four species to the low light condition.
3. The added information in the legend of Figure 3 is not sufficient and English is awkward. Especially the information about the meaning of the thin arrows and dashed lines are scarce. If authors did not use any statistical method for the thin arrows or dashed lines in this figure, I think this figure should be deleted.
Author Response
Dear reviewer
Re: Manuscript ID: plants-1377753
Please find attached a revised version of our manuscript “The effect of low irradiance on leaf nitrogen allocation and mesophyll conductance to CO2 in seedlings of four tree species in subtropical China”, which we would like to resubmit for publication as a research paper in Plants.
Your comments were highly insightful and enabled us to greatly improve the quality of our manuscript. In the following pages are our point-by-point responses to each of the comments of the reviewers as well as your own comments.
Revisions in the text are shown using yellow highlight for additions, and strikethrough font [example] for deletions. We hope that the revisions in the manuscript and our accompanying responses will be sufficient to make our manuscript suitable for publication in Plants.
We shall look forward to hearing from you at your earliest convenience.
Thank you and best regards.
Yours sincerely,
Jingchao Tang
Ph: (+86)18866482400
E–mail: tjc@qut.edu.cn
Responses to the comments of Reviewer #2
Question 1: Authors used the photosynthetic data under PPFD ≤ 200 umol m-2 s-1 for the calculation of apparent quantum yield (AQY). However, the light intensity would be too strong that the photosynthetic rate could not be fitted by linear fitting (as we can see in Fig. S1). Because authors measured photosynthetic rate at 0, 10, 20, 30, 50 umol m-2 s-1 and more light intensities, the photosynthetic data under PPFD ≤ 30 or 50 umol m-2 s-1 would be enough for the statistically robust calculation of AQY. The results will be largely changed.
Response: You are right, and we changed to use the photosynthetic data under PPFD ≤ 30 umol m-2 s-1 for the calculation of AQY, and our data were largely changed. We also rewritten “Discussion” and “Materials and Methods” section about AQY in our manuscript.
[Line 178 to 182 in revised version]
Table 5. Apparent quantum yield (AQY), dark respiration (Rn), light compensation point (LCP) and light saturation point (LSP) in Dalbergia odorifera, Erythrophleum fordii, Betula alnoides, and Castanopsis hystrix grown under three different irradiance treatments. Data are means of seven plants per treatment ± SE. Lower case letters indicate significant difference at 0.05 levels among the irradiance treatments, whereas capital letters indicate significant difference at 0.05 levels among species under same irradiance treatment. F-ratios with statistically significant values denoted by *P < 0.05, **P < 0.01, ***P < 0.001 among irradiance treatment.
Tree species |
Irradiance treatment |
AQY (mol mol–1) |
Rn (μmol m–2 s–1) |
LCP(μmol m–2 s–1) |
LSP(μmol m–2 s–1) |
Dalbergia Odorifera |
100% |
0.052 ± 0.004aA |
0.909 ± 0.050aBC |
22.1 ± 1.68aA |
822.9 ± 27.5aA |
40% |
0.059 ± 0.002aA |
0.845 ± 0.050aA |
14.4 ± 0.73bB |
724.3 ± 37.0abA |
|
10% |
0.058 ± 0.002aA |
0.760 ± 0.038aB |
7.4 ± 0.43cB |
684.3 ± 23.1bA |
|
F |
2.300 |
2.587 |
46.127*** |
5.378* |
|
Erythrophleum fordii |
100% |
0.047 ± 0.003bA |
1.129 ± 0.051aA |
13.9 ± 0.81aB |
637.1 ± 29.6aB |
40% |
0.062 ± 0.002aA |
0.873 ± 0.050bA |
13.1 ± 1.10aB |
633.6 ± 17.1aA |
|
10% |
0.059 ± 0.001aA |
0.936 ± 0.030bAB |
7.5 ± 0.79bB |
522.1 ± 17.2bB |
|
F |
15.924*** |
8.811** |
14.464*** |
6.569** |
|
Castanopsis hystrix |
100% |
0.047 ± 0.003aA |
1.005 ± 0.067aAB |
14.0 ± 1.21aB |
632.9 ± 23.4aB |
40% |
0.054 ± 0.002aA |
0.988 ± 0.040aA |
7.3 ± 1.00bC |
307.1 ± 26.8bC |
|
10% |
0.054 ± 0.003aA |
1.048 ± 0.088aA |
7.4 ± 1.21bB |
262.1 ± 27.7bC |
|
F |
2.737 |
0.206 |
11.226** |
60.359*** |
|
Betula alnoides |
100% |
0.049 ± 0.001aA |
0.889 ± 0.039aBC |
20.8 ± 0.93aA |
886.4 ± 43.5aA |
40% |
0.055 ± 0.003aA |
0.844 ± 0.055aA |
18.6 ± 2.49aA |
519.3 ± 27.9bB |
|
10% |
0.051 ± 0.003aA |
0.834 ± 0.048aAB |
10.5 ± 0.94bA |
268.6 ± 30.5cC |
|
F |
1.415 |
0.384 |
10.989** |
80.343*** |
[Line 291 to 293 in revised version]
AQY refers to the ability to use low light [43]. E. Fordii exhibited improved AQY under the 10% and 40% irradiance treatments, and the AQY values under the 10% irradiance treatment in the order of: E. fordii > D. odorifera > C. hystrix > B. Alnoides (Table 5). and also reduced Rn under the 10% and 40% irradiance treatments to reduce respiratory expenditure (Table 5).
[Line 359 to 360 in revised version]
The AQY (mol mol–1) was measured as the initial slope of the light response curves (PPFD ≤ 200 30 μmol m–2 s–1).
Question 2: As authors mentioned in line 186, this study discusses about the PNUE under low-irradiance treatment in many places. However, as I mentioned in the previous comments, PNUE in this study only uses Asat (photosynthesis rate at saturating light intensity) for the calculation, which does not indicate the adaptability of the plants to low light condition. (Authors currently just using the data of PNUE at saturating light intensity of plants grown under low light: 10% light condition.) Authors should use, for example, photosynthesis rate at 100 umol m-2 s-1 (which is in the range of the growth irradiance in the 10% light condition), for the calculation of PNUE (which can be called as PNUE100 and so on). Otherwise, the almost all analyses about PNUE in the current manuscript does not indicate the adaptability of the four species to the low light condition.
Response: You are right, and we added data of A100 (Net CO2 assimilation rate at PPFD of 100 umol m-2 s-1), A400 (Net CO2 assimilation rate at PPFD of 400 umol m-2 s-1), PNUE100 (A100/Narea×14) and PNUE400 (A400/Narea×14) to indicate the adaptability of the four species to the low light condition. We added a table (new Table 6) to show data of these traits, and described and discussed these traits. We also changed former PNUE (Asat/Narea×14) to PNUEsat all through our manuscript.
[Line 195 to 200 in revised version]
Table 6. Net CO2 assimilation rate at PPFD of 100 umol m-2 s-1 (A100); net CO2 assimilation rate at PPFD of 400 umol m-2 s-1 (A400); photosynthetic N use efficiency at PPFD of 100 umol m-2 s-1 (PNUE100); photosynthetic N use efficiency at PPFD of 400 umol m-2 s-1 (PNUE400) in Dalbergia odorifera, Erythrophleum fordii, Betula alnoides, and Castanopsis hystrix grown under three different irradiance treatments. Data are means of seven plants per treatment ± SE. Lower case letters indicate significant difference at 0.05 levels among the irradiance treatments, whereas capital letters indicate significant difference at 0.05 levels among species under same irradiance treatment. F-ratios with statistically significant values denoted by *P < 0.05, **P < 0.01, ***P < 0.001 among irradiance treatment.
Tree species |
Irradiance treatment |
A100 (μmol m–2 s–1) |
A400 (μmol m–2 s–1) |
PNUE100 (μmol mol–1 s–1) |
PNUE400 (μmol mol–1 s–1) |
Dalbergia Odorifera |
100% |
2.78±0.41bA |
7.38±0.45abAB |
20.7±1.36cB |
48.2±3.65cB |
40% |
4.06±0.08aA |
8.02±0.30aA |
35.2±0.66bB |
69.8±3.35bB |
|
10% |
4.03±0.17aA |
6.65±0.36bA |
59.2±4.12aAB |
97.4±7.32aA |
|
F |
7.114* |
6.206* |
58.81*** |
23.261*** |
|
Erythrophleum fordii |
100% |
3.42±0.16bA |
6.07±0.24bB |
24.2±1.38bB |
42.7±1.48cB |
40% |
4.04±0.08aA |
8.53±0.22aA |
32.3±0.72aB |
68.4±2.42aB |
|
10% |
3.98±0.18aA |
6.31±0.43bA |
35.8±1.29aC |
56.6±3.11bB |
|
F |
5.426* |
16.016*** |
26.501*** |
22.155*** |
|
Castanopsis hystrix |
100% |
3.57±0.12aA |
7.67±0.42aAB |
49.1±1.83aA |
105.4±6.18aA |
40% |
3.00±0.15bB |
3.98±0.29bB |
57.3±4.57aA |
76.2±8.00bB |
|
10% |
2.92±0.19bB |
3.68±0.31bB |
51.8±3.22aBC |
65.3±4.99bB |
|
F |
5.107* |
41.202*** |
1.506 |
10.123** |
|
Betula alnoides |
100% |
3.27±0.15aA |
7.95±0.52aA |
46.4±4.09bA |
113.7±12.66aA |
40% |
3.52±0.28aAB |
7.16±0.68aA |
67.3±7.62abA |
135.9±14.91aA |
|
10% |
2.92±0.32aB |
3.91±0.66bB |
72.9±6.33aA |
95.6±11.85aA |
|
F |
1.358 |
12.425*** |
5.074* |
2.852 |
[Line 184 to 193 in revised version]
A100 and A400 in D. odorifera and E. fordii seedling leaves were significantly higher than those in C. hystrix and B. alnoides under 10% irradiance treatment (Table 6). A100, PNUE100 and PNUE400 of D. odorifera and E. fordii under the 10% and 40% irradiance treatments were significantly higher than that under the 100% treatment (Table 6). A400 of E. fordii under the 40% irradiance treatment was significantly higher than that under the other treatments (Table 6). A100, A400 and PNUE400 of C. hystrix under the 10% and 40% irradiance treatments were significantly lower than that under the 100% treatment (Table 2). A400 of B. alnoides was significantly lower than that under the 40% and 100% treatments, but PNUE100 of B. alnoides was significantly higher than that under the 100% treatment (Table 6).
[Line 280 to 291 in revised version]
As these species are commonly used to plant in gaps or mixed with other species, their tolerance to low light conditions will affect the growth effect after planting. All four species decreased the LMA to increase the area receiving light (Table 1) [1,4,8,14], increased PL to increase their light-harvesting capacity and sustain photosynthesis (Table 4) [3–4], and decreased the LCP and LSP to increase the ability to use low light (Table 5) under low light conditions. Meanwhile, N-fixing plants exhibited some other adaptations to low light conditions, such as increased Ci and Cc, maintained Asat under the 10% and 40% irradiance treatments (Tables 1 and 2), and increased PNUE under the 10% and 40% irradiance treatments (Table 1). Meanwhile, N-fixing plants exhibited some other adaptations to low light conditions, such as increased A100, PNUE100 and PNUE400 under the 10% and 40% irradiance treatments (Table 6). N-fixing plants also had higher A100, A400, Narea, Vcmax and Jmax than non-N-fixing species under the 10% irradiance treatment (Table 1,3,6). Overall, these two N-fixing plant seedlings had higher photosynthetic rates, photosynthetic ability and higher adjustment ability of photosynthetic N use under low light conditions. AQY refers to the ability to use low light [43].
[Line 352 to 360 in revised version]
were measured at photon flux densities of 1500, 1200, 1000, 800, 600, 400, 200, 150, 100, 80, 50, 30, 20, 10, and 0 μmol m–2 s–1 (see Fig. S1). The PPFD-saturated net CO2 assimilation rate (Asat, μmol m–2 s–1), net CO2 assimilation rate at PPFD of 100 umol m-2 s-1 (A100), net CO2 assimilation rate at PPFD of 400 umol m-2 s-1 (A400), dark respiration (Rn, μmol m–2 s–1), LSP (μmol m–2 s–1), and LCP (μmol m–2 s–1) were then measured from the light response curves. (100 and 400 umol m-2 s-1 were in the range of the growth irradiance in the 10% and 40% light conditions, respectively). The AQY (mol mol–1) was measured as the initial slope of the light response curves (PPFD ≤ 30 μmol m–2 s–1).
[Line 388 to 391 in revised version]
while the PNUE (μmol mol–1 s–1) was calculated as:
......................................................(2)
where PNUEsat was calculated by Asat and Narea, PNUE100 was calculated by A100 and Narea and PNUE400 was calculated by A400 and Narea, respectively.
Question 3: The added information in the legend of Figure 3 is not sufficient and English is awkward. Especially the information about the meaning of the thin arrows and dashed lines are scarce. If authors did not use any statistical method for the thin arrows or dashed lines in this figure, I think this figure should be deleted..
Response: I agree with your perspective, and deleted Fig.3 and discussion about Fig.3 in our manuscript.
[Line 306 to 308 in revised version]
To better characterize the changes occurring in the four tree species under low light treatment, we illustrated the variation of each characteristic under the 10% irradiance treatment (Fig. 3). The increases in PR, PB, and PL increased the PNUE in D. odorifera seedling leaves under the 10% irradiance treatment; the increases in PR and PL increased the PNUE in E. fordii seedling leaves; the decreases in PR and PB decreased the PNUE in C. hystrix seedling leaves, which offset the influence of increased PL, and no significant changes in PR and PB resulted in a stable PNUE in B. alnoides seedling leaves (Table 4, Fig. 3). The PL of all four species increased to improve their light-trapping ability under low irradiance treatments (Table 4), which was consistent with previous studies [31,46].
[Line 306 to 308 in revised version]
Fig. 3. Leaf trait changes in Dalbergia odorifera, Erythrophleum fordii, Betula alnoides, and Castanopsis hystrix grown under 10% irradiance versus 100% irradiance treatments. “Red upward arrow” means this leaf trait under 10% irradiance treatment was significant higher than 100%, “Blue down arrow” means this leaf trait under 10% irradiance treatment was significant lower than 100%, and “green rectangle” means this leaf trait under 10% irradiance treatment was no significant change with 100%. Specific data information and significance see Tables 1-5. The leaf trait pointed at by “Thin blue arrow” can be affected by the traits at the other end of arrow, and the increased part of leaf trait pointed at by “Thin red arrow” might come from the reduced part of leaf traits at the other end of arrow.
